# YOD1 sustains NOD2-mediated protective signaling in colitis by stabilizing RIPK2

Jiangyun Shen [1], Liyan Lou[1], Xue Du[1], Bincheng Zhou [1], Yanqi Xu[1], Fuqi Mei[1], Liangrong Wu[1,2], Jianmin Li[3], Ari Waisman [4], Jing Ruan [3✉] & Xu Wang [1✉]

## Abstract

**Inflammatory bowel disease (IBD) is a disorder causing chronic inflammation in the gastrointestinal tract, and its pathophysiological mechanisms are still under investigation. Here, we find that mice deficient of YOD1, a deubiquitinating enzyme, are highly susceptible to dextran sulfate sodium (DSS)-induced colitis. The bone marrow transplantation experiment reveals that YOD1 derived from hematopoietic cells inhibits DSS colitis. Moreover, YOD1 exerts its protective role by promoting nucleotide-binding oligomerization domain 2 (NOD2)-mediated physiological inflammation in macrophages. Mechanistically, YOD1 inhibits the proteasomal degradation of receptor-interacting serine/threonine kinase 2 (RIPK2) by reducing its K48 polyubiquitination, thereby increasing RIPK2 abundance to enhance NOD2 signaling. Consistently, the protective function of muramyldipeptide, a NOD2 ligand, in experimental colitis is abolished in mice deficient of YOD1. Importantly, YOD1 is upregulated in colon-infiltrating macrophages in patients with colitis. Collectively, this study identifies YOD1 as a novel regulator of colitis.**

**Keywords** Inflammatory Bowel Disease; NOD2; RIPK2; YOD1; Ubiquitination
**Subject Categories** Immunology; Post-translational Modifications & Proteolysis; Signal Transduction

## Introduction

Inflammatory bowel disease (IBD), represented by ulcerative colitis (UC) and Crohn's disease (CD), causes chronic inflammation and severe tissue damage in the gastrointestinal tract, generating debilitating symptoms like abdominal pain, diarrhea, bloody stool, fever, and weight loss (Kobayashi et al, 2020; Peloquin et al, 2016; Roda et al, 2020). Although IBD is a multifactorial disorder resulting from a complicated interplay among genetic, environmental and immunological factors, an abnormal inflammatory response to intestinal microbiota in a genetically susceptible host is considered to be a pivotal process underlying the pathogenesis of IBD (Cohen et al, 2019; Ruan et al, 2022).

Inflammatory responses are tightly regulated to efficiently restore tissue homeostasis while averting immunopathology. In recent years, ubiquitination, a type of post-translational modification, has emerged as a crucial mechanism regulating inflammatory activation (Ruan et al, 2022). Ubiquitination is sequentially catalyzed by E1 ubiquitin-activating enzymes, E2 ubiquitin-conjugating enzymes, and E3 ubiquitin ligases. Besides, deubiquitinating enzymes (DUBs) counter-regulate ubiquitination, enabling the precise modulation of inflammatory responses (Clague Urbe and Komander, 2019; Rape, 2018). Interestingly, accumulative evidence has stressed the importance of DUBs in IBD (Ruan et al, 2022). The NF-κB pathway plays a key role in colonic inflammation, and it is activated by components of the luminal bacteria, such as lipopolysaccharide (LPS) and peptidoglycan (PGN), as well as IBD-associated detrimental cytokines like TNF-α (Cohen and Sachar, 2017; D'Haens and van Deventer, 2021; Zhang Lenardo and Baltimore, 2017). The DUB A20 serves as an important post-translational brake on the NF-κB signaling, and specific deletion of A20 in macrophages, intestinal epithelial cells (IECs), or dendritic cells predisposes mice to colitis (Hammer et al, 2011; Pu et al, 2021; Vereecke et al, 2010; Vereecke et al, 2014). Similar to A20, OTUD1 is also a DUB of the ovarian tumor protease (OTU) subfamily. OTUD1 has been shown to ameliorate experimental colitis induced by dextran sulfate sodium (DSS) via inhibiting RIPK1-mediated NF-κB activation (Wu et al, 2022).

YOD1, also known as OTUD2, is another DUB of the OTU subfamily that can influence the NF-κB signaling (Ernst et al, 2009; Schimmack et al, 2017). YOD1 has been shown to inhibit IL-1-induced NF-κB activation by competitively associating with TRAF6 through a non-catalytic mechanism (Schimmack et al, 2017). Besides, utilizing its catalytic activity, YOD1 inhibits antiviral innate immune responses by K63 deubiquitinating MAVS (Liu et al, 2019). In addition to these non-degradative activities, YOD1 can also deubiquitinate several proteins to enhance their stability. YOD1 regulates the Hippo signaling pathway by deubiquitinating and stabilizing ITCH (Kim et al, 2017). Two recent studies show that YOD1 affects head and neck squamous cell carcinoma and triple-negative breast cancer by stabilizing TRIM33 and CDK1,

[1]Oujiang Laboratory (Zhejiang Lab for Regenerative Medicine, Vision and Brain Health); School of Pharmaceutical Sciences, Wenzhou Medical University, 325035 Wenzhou, China. [2]Department of Pharmacy, Yiwu Central Hospital, 322099 Yiwu, China. [3]Department of Pathology, The First Affiliated Hospital, Wenzhou Medical University, 325000 Wenzhou, China. [4]Institute for Molecular Medicine, Johannes Gutenberg University Mainz, 55131 Mainz, Germany. ✉E-mail: ruanjing850617@163.com; sunrim@163.com

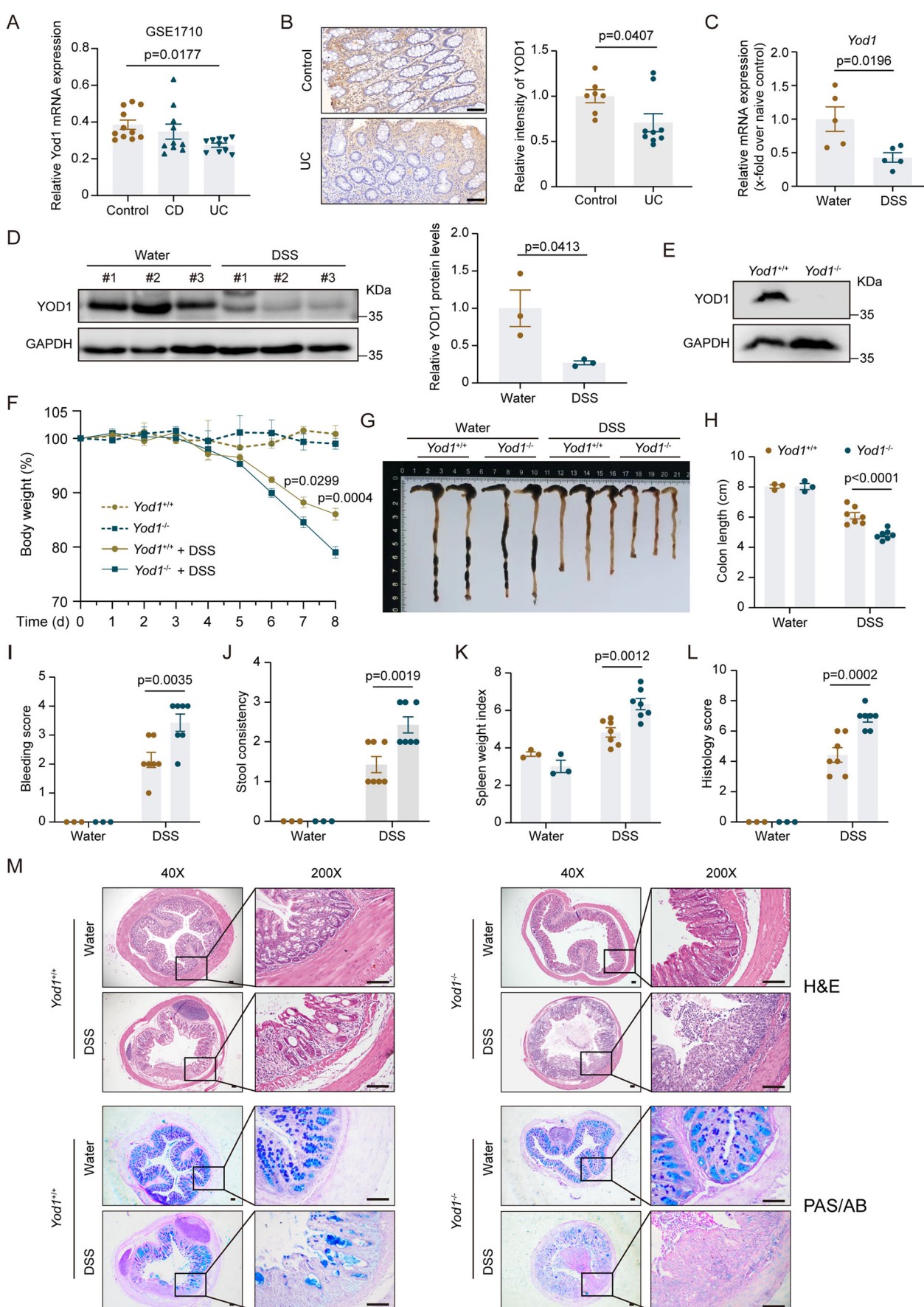

**Figure 1. Yod1$^{-/-}$ mice are susceptible to DSS-induced colitis.**

(A) Expression of Yod1 mRNA in colons of normal controls (n = 11, biological replicates), patients with CD (n = 10, biological replicates), and patients with UC (n = 10, biological replicates) based on microarray data (GSE1710). (B) Representative immunohistochemical staining (left) and quantification (right) of YOD1 in control (n = 7, biological replicates) and UC (n = 9, biological replicates) colon samples. Scale bar = 100 μm. (C) Relative levels of Yod1 mRNA in the colon of control mice and 2.5% DSS-treated mice on day 7 were determined by qRT-PCR (n = 5, biological replicates). (D) YOD1 protein levels in the colon of control mice and 2.5% DSS-treated mice on day 7 were analyzed by western blot (left). The right panel shows the relative quantification (n = 3/group, biological replicates). (E) Western blot analysis of YOD1 protein abundance in the colon of Yod1$^{+/+}$ and Yod1$^{-/-}$ mice. (F) Body weight change of Yod1$^{+/+}$ and Yod1$^{-/-}$ mice challenged with 2.5% DSS for 5 days followed by water for 3 days (n = 7, biological replicates). (G, H) Representative image (G) and length (H) of colons of Yod1$^{+/+}$ and Yod1$^{-/-}$ mice on day 8 after DSS treatment (n = 3–7, biological replicates). (I–L) Rectal bleeding (I), stool consistency (J), spleen weight (K), and colon histology score (L) of Yod1$^{+/+}$ and Yod1$^{-/-}$ mice on day 8 after DSS treatment (n = 3–7, biological replicates). (M) Representative H&E and PAS/AB staining of colons from Yod1$^{+/+}$ and Yod1$^{-/-}$ mice on day 8 after DSS treatment. Original magnification, ×40 and ×200. Scale bar = 100 μm. Data information: Data in (B–M) are representative of three replicates. Data in (A–D, F, H–L) show the mean ± SEM. Statistical analyses were performed using one-way ANOVA followed by the Dunnett post-test (A), two-tailed unpaired Student's t test (B–D), and two-way ANOVA followed by the Sidak post-test (F, H–L). Source data are available online for this figure.

respectively (Han et al, 2023; Wu et al, 2023). Moreover, YOD1 plays a role in protein quality control by regulating endoplasmic reticulum-associated protein degradation (Ernst et al, 2009; Sasset et al, 2015). However, the in vivo function of YOD1 remains largely unknown.

In this study, we found that YOD1 expression was downregulated in colons of both UC patients and DSS-treated mice. To study the role of YOD1 in colitis, we generated Yod1$^{-/-}$ mice and found that ablation of YOD1 aggravated DSS-induced colitis. Deeper investigation of the functionality revealed that YOD1 ameliorated colonic damage by bolstering nucleotide-binding oligomerization domain 2 (NOD2)-mediated protective effects in macrophages. Mechanistically, YOD1 enhances NOD2-mediated signal transduction by stabilizing receptor-interacting serine/threonine kinase 2 (RIPK2). Collectively, the findings of this study identified YOD1 as a novel regulator of NOD2 signaling and experimental colitis, and highlight the potential of YOD1 as a beneficial therapeutic target for IBD.

## Results

### Ablation of YOD1 aggravates experimental colitis in mice

To assess whether YOD1 plays a role in IBD, we first analyzed the expression of YOD1 in clinical samples collected from normal controls and patients with CD or UC. At both mRNA and protein levels, YOD1 expression was markedly reduced in the colon tissue of UC patients as compared with normal controls (Fig. 1A,B). Similarly, mRNA and protein levels of YOD1 were also significantly reduced in the colon tissue of DSS-treated mice (Fig. 1C,D; Appendix Fig. S1), indicative of a potential role of YOD1 in colitis. To investigate the functional role of YOD1 in colitis, we generated Yod1$^{-/-}$ mice (Fig. 1E; Appendix Fig. S2A). The Yod1$^{-/-}$ mice were born normally and reached adulthood without obvious defects. Besides, they displayed normal intestinal histology until 12 months after birth (Appendix Fig. S2B). However, upon DSS treatment, Yod1$^{-/-}$ mice lost significantly more body weight in comparison to Yod1$^{+/+}$ mice (Fig. 1F; Appendix Fig. S3). In addition, as compared with control mice, Yod1$^{-/-}$ mice exhibited aggravated colitis symptoms including reduced colon length, increased rectal bleeding, exacerbated diarrhea, and enhanced splenomegaly in response to DSS treatment (Fig. 1G–K). Further histopathological examination found that Yod1$^{-/-}$ mice had exacerbated damage in the colon upon DSS challenge, as manifested by extensive epithelial

erosion, widespread leukocyte infiltration, and more ulcerations (Fig. 1L,M). Consistently, PAS/AB staining showed that fewer goblet cells were maintained in Yod1$^{-/-}$ mice during DSS-induced colitis (Fig. 1M; Appendix Fig. S4). Together, these results show that YOD1 deficiency aggravates colonic injury in experimental colitis.

During IBD, the gastrointestinal tract is damaged by excessive proinflammatory cytokines, which are mainly secreted by infiltrating leukocytes. Therapies against proinflammatory cytokines have been proven to be clinically efficacious in IBD treatment. Although the basal levels of cytokines and chemokines in the colon were comparable between Yod1$^{+/+}$ and Yod1$^{-/-}$ mice, their expression was significantly elevated in Yod1$^{-/-}$ mice as compared with Yod1$^{+/+}$ mice upon DSS challenge (Fig. 2A–F). Macrophages are a major source of proinflammatory cytokines and chemokines, and we then analyzed the colonic infiltration of macrophages by immunohistochemistry and flow cytometry. In line with the expression pattern of cytokines and chemokines shown in Fig. 2A–F, the colonic infiltration of macrophages was prominently increased in Yod1$^{-/-}$ mice as compared with control mice after DSS treatment (Fig. 2G–I; Appendix Fig. S5A,B). Taken together, these findings demonstrate that ablation of YOD1 aggravates experimental colitis in mice, indicative of a protective role of YOD1 in IBD.

### Deletion of YOD1 in hematopoietic cells aggravates DSS-induced colitis

Given that (i) proinflammatory cytokines cause tissue damage in colitis, (ii) proinflammatory cytokines are predominantly produced by macrophages in DSS-induced colitis, and (iii) ablation of YOD1 increased the expression of proinflammatory cytokines and chemokines in mouse colons upon DSS challenge, it is highly possible that YOD1 influences colitis by regulating macrophages. Interestingly, in sharp contrast to the finding that colitis induced the overall downregulation of YOD1 in colon tissue, YOD1 expression was upregulated in colon-infiltrating macrophages in both mice and humans during colitis (Fig. 3A,B; Appendix Fig. S6). To confirm the role of macrophage-derived YOD1, we generated bone marrow chimeric mice with YOD1-sufficient and -deficient macrophages by reconstituting irradiated C57/BL6 mice with bone marrow from Yod1$^{+/+}$ and Yod1$^{-/-}$ mice, respectively (Appendix Fig. S7). As compared with mice transplanted with YOD1-sufficient bone marrow (WT → WT), YOD1 was efficiently reduced in the splenocytes of mice receiving YOD1-deficient bone marrow (KO → WT) (Fig. 3C). Upon DSS stimulation, the KO → WT mice

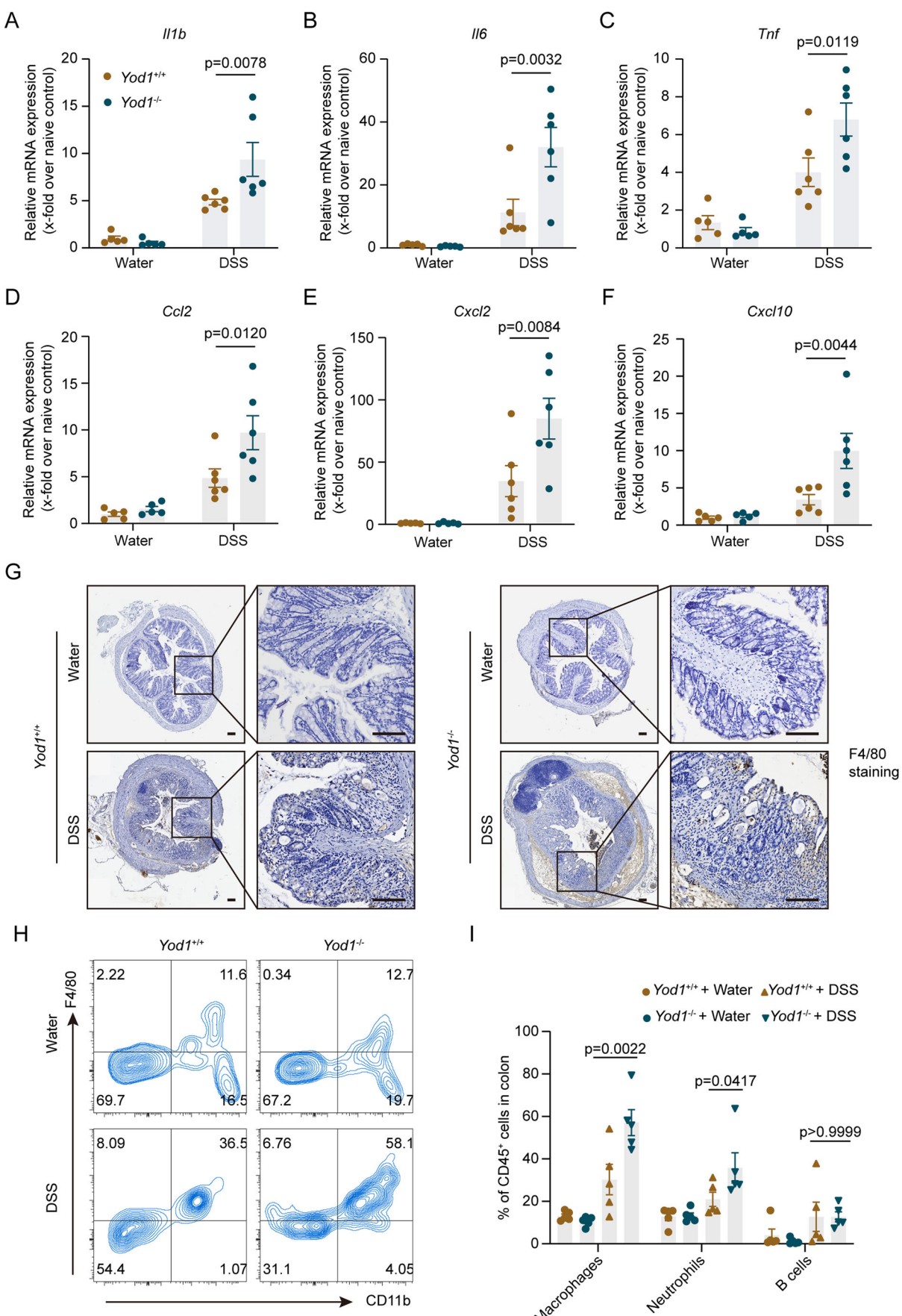

**Figure 2. Increased proinflammatory gene transcription and macrophage infiltration in the colon of Yod1⁻/⁻ mice during colitis.**

(A–F) The relative expression of *Ilb* (A), *Il6* (B), *Tnf* (C), *Ccl2* (D), *Cxcl2* (E), and *Cxcl10* (F) mRNA in colons of *Yod1⁺/⁺* and *Yod1⁻/⁻* mice on day 8 after DSS treatment was determined by qRT-PCR (n = 5–6, biological replicates). (G) Representative F4/80 immunohistochemistry staining of colons from *Yod1⁺/⁺* and *Yod1⁻/⁻* mice on day 8 after DSS treatment. Original magnification, ×40 and ×200. Scale bar = 100 μm. (H, I) Colon-infiltrating leukocytes were analyzed by flow cytometry. Representative dot plots (H) and quantification (I) of macrophages are shown (n = 5, biological replicates). Data information: Data in (A–G) are representative of three replicates, and data in (H, I) are representative of two replicates. Data in (A–F, I) show the mean ± SEM. Statistical analyses were performed using two-way ANOVA followed by the Sidak post-test (A–F, I). Source data are available online for this figure.

lost more body weight than did the WT → WT counterparts (Fig. 3D). Consistently, colitis-induced rectal bleeding, diarrhea, and colonic shortening was also more prominent in the KO → WT group (Fig. 3E–H). Besides, the KO → WT mice had more severe intestinal damage and lost more goblet cells than did the WT → WT mice, as evidenced by the H&E and PAS/AB staining (Fig. 3I,J, Appendix Fig. S8). In addition to macrophages, IECs are also key players in colitis and they form the first defense line in DSS-induced colitis. To study the role of YOD1 in IECs, we generated *Yod1⁻/⁻* Mode-K cells using the CRISPR/Cas9 technology (Appendix Fig. S9A). Upon stimulation with CHX and TNF-α, a detrimental proinflammatory cytokine underlying mucosal damage in IBD, cell viability was comparable between YOD1-deficient and -sufficient Mode-K cells (Appendix Fig. S9B), decreasing the possibility that YOD1 affects colitis by regulating the apoptosis of IECs. In aggregate, these results demonstrate that hematopoietic cell-derived YOD1 plays a key role in DSS-induced colitis.

## YOD1 deficiency diminishes MDP-induced gene expression in macrophages

In the intestine, macrophages recognize luminal microbes through pattern recognition receptors (PRRs) including Nod-like receptors (NLRs) and Toll-like receptors (TLRs). NOD2, a cytoplasmic NLR, recognizes the PGN component MDP and induces physiological inflammation to clear the bacteria. Clinically, loss-of-function mutations of NOD2 are linked with IBD (Hugot et al, 2001; Ogura et al, 2001). We then assessed the impact of YOD1 on NOD2-mediated inflammatory responses by stimulating YOD1-sufficient and -deficient BMDMs with L18-MDP, a lipophilic derivative of MDP. As shown in Fig. 4A–F, deletion of YOD1 significantly reduced L18-MDP-induced expression of cytokines and chemokines in macrophages. In addition to NOD2, TLR4 is another key PRR applied by macrophages to detect and respond to microbes. However, YOD1 deletion had no impact on the expression of cytokines induced by lipopolysaccharide (LPS), a specific TLR4 ligand (Appendix Fig. S10A–C). Moreover, YOD1 deficiency did not change cytokine production and signal transduction in macrophages upon stimulation with TNF, a detrimental cytokine in colitis (Appendix Fig. S10C–F). Collectively, these findings show that YOD1 critically regulates NOD2-mediated inflammatory responses in macrophages.

## YOD1 sustains NOD2 signaling by stabilizing RIPK2

To decipher the molecular mechanism underlying the regulatory role of YOD1 on NOD2-mediated inflammatory responses, we analyzed the impact of YOD1 on L18-MDP-induced signaling by western blot. Consistent with the results in Fig. 4A–F, L18-MDP-

induced activation of NF-κB and MAPK signaling in BMDMs was strongly diminished by YOD1 deletion (Fig. 5A). As a DUB, YOD1 regulates signal transduction by deubiquitinating protein substrates. To identify the protein substrates directly regulated by YOD1, we harvested YOD1 and its interacting proteins from RAW264.7 cells by immunoprecipitation and analyzed the immunocomplex by mass spectrometry. Among the identified proteins in the immunocomplex, RIPK2 is a key component of the NOD2 signaling (Fig. 5B; Datasets EV1–3). Upon activation, NOD2 undergoes self-oligomerization and interacts with RIPK2, which then recruits TAK1 to activate downstream NF-κB and MAPK pathways (Ruan et al, 2022). Intriguingly, the protein abundance of RIPK2 was strongly reduced in the absence of YOD1 (Fig. 5C). However, YOD1 deletion did not decrease the protein levels of NOD2 and XIAP, an E3 ligase critical for NOD2:RIPK2 signaling (Fig. 5C). Moreover, the interaction of RIPK2 and XIAP was not affected by YOD1 deficiency (Fig. 5C).

Further analysis confirmed that YOD1 ablation significantly reduced the protein levels of RIPK2 in BMDMs (Fig. 5C–E). Similarly, as compared with *Yod1⁺/⁺* mice, *Yod1⁻/⁻* mice had significantly less RIPK2 protein in colons (Appendix Fig. S11A). In contrast, YOD1 deletion had no influence on RIPK2 transcription (Fig. 5F), ruling out the possibility that YOD1 increases RIPK2 protein levels by enhancing its de novo synthesis. As a DUB, YOD1 has been shown to increase the stability of various protein substrates. Indeed, we found that, after blocking protein synthesis with CHX, YOD1 deletion accelerated the degradation of RIPK2 in BMDMs in the presence or absence of L18-MDP (Fig. 5G; Appendix Fig. S12A), indicating that YOD1 increases the stability of RIPK2 protein. Consistently, YOD1 overexpression increased RIPK2 protein abundance and delayed RIPK2 degradation upon CHX treatment (Appendix Fig. S12B). In line with the western blot results, immunofluorescence staining also found that RIPK2 stability was enhanced in YOD1-overexpressed macrophages (Fig. 5H). Since RIPK2 is essential for NOD2 signaling and YOD1 stabilizes RIPK2, overexpression of YOD1 strongly increased L18-MDP-induced NF-κB and MAPK activation as well as cytokine production (Appendix Fig. S13). Altogether, these findings show that YOD1 enhances NOD2 signaling in macrophages by stabilizing RIPK2.

## YOD1 binds and K48 deubiquitinates RIPK2

Given that RIPK2 was identified as a YOD1-interacting protein by mass spectrometry (Fig. 5B) and that YOD1 critically regulated RIPK2 stability (Fig. 5D–H; Appendix Fig. S12), we assessed the direct interaction between YOD1 and RIPK2 by co-immunoprecipitation and western blot. YOD1 was found to directly bind RIPK2 in both endogenous and exogenous systems (Fig. 6A,B). Consistently, immunofluorescence staining

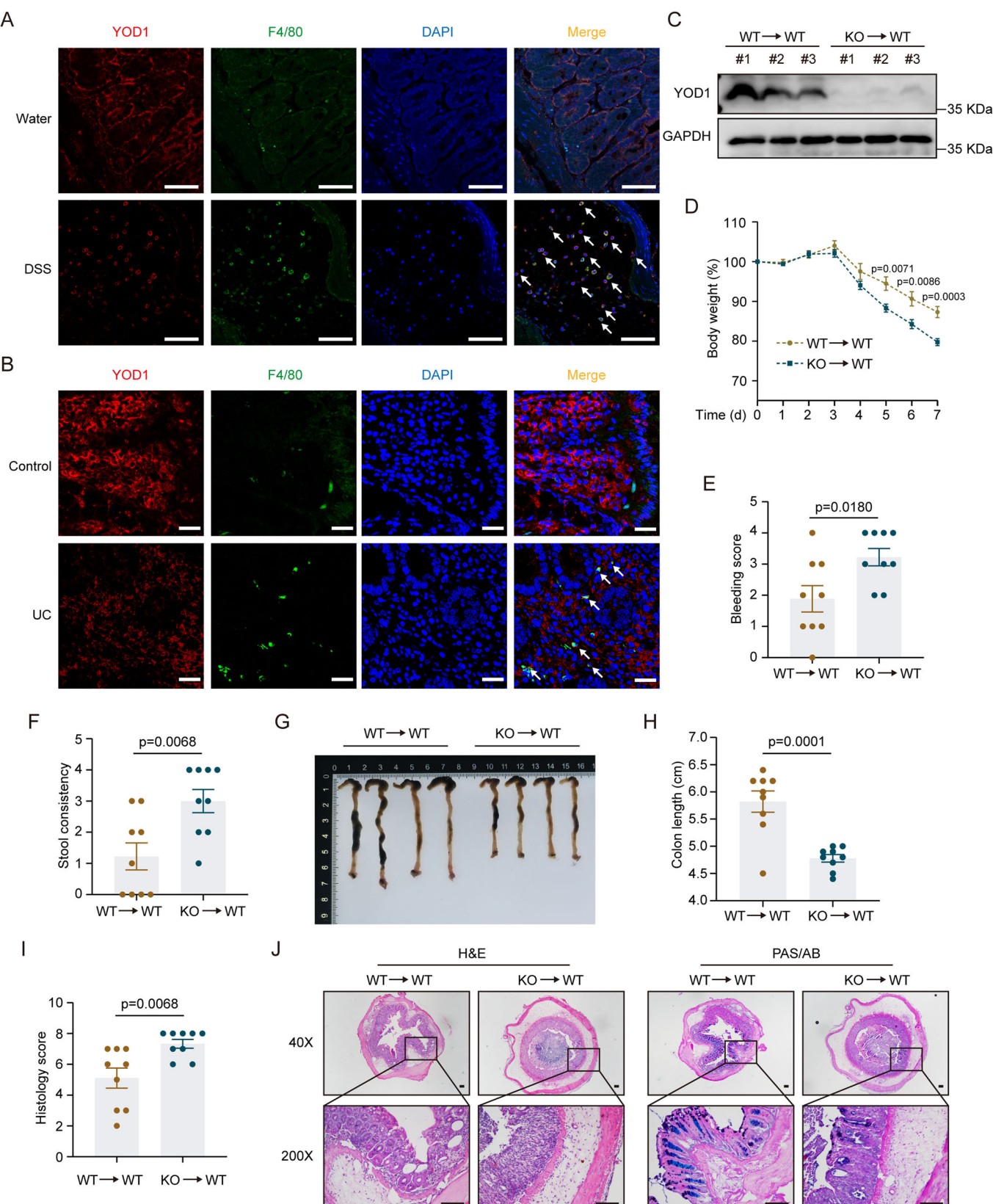

**Figure 3. YOD1 is upregulated in colon-infiltrating macrophages during colitis and pan-hematopoietic deletion of YOD1 exacerbates colitis.**

(A) Representative YOD1 (red) and F4/80 (green) immunofluorescence staining of the colon from control and DSS-treated mice on day 8. Original magnification, ×200. Scale bar = 100 μm. (B) Representative immunofluorescence staining of YOD1 (red) and F4/80 (green) in control and UC colon samples. Original magnification, ×200. Scale bar = 25 μm. (C) Western blot analysis of YOD1 expression in splenocytes of bone marrow chimeric mice. (D) Body weight change of bone marrow chimeric mice challenged with 2.5% DSS for 5 days followed by water for 2 days (*n* = 9, biological replicates). (E, F) Rectal bleeding (E) and stool consistency (F) of bone marrow chimeric mice on day 7 after DSS treatment (*n* = 9, biological replicates). (G, H) Representative image (G) and length (H) of colons from bone marrow chimeric mice on day 7 after DSS treatment (*n* = 9, biological replicates). (I) Colon histology score of bone marrow chimeric mice on day 7 after DSS treatment (*n* = 9). (J) Representative H&E and PAS/AB staining of colons from bone marrow chimeric mice on day 7 after DSS treatment. Original magnification, ×40 and ×200. Scale bar = 100 μm. Data information: Data in (A, B) are representative of three replicates, and data in (C–J) are representative of two replicates. Data in (D–F, H, I) show the mean ± SEM. Statistical analyses were performed using two-tailed unpaired Student's *t* test (D–F, H, I). Source data are available online for this figure.

demonstrated that YOD1 co-localized with RIPK2 in the cytoplasm of macrophages (Fig. 6C). In addition, stimulation with L18-MDP, rather than LPS and TNF, further increased the binding between YOD1 and RIPK2 (Appendix Fig. S14). Considering that (i) YOD1 reduced the degradation of RIPK2 (Fig. 5D–H; Appendix Fig. S12), (ii) K48 polyubiquitination mediates the proteasomal degradation of proteins, and (iii) YOD1 is a DUB that shows linkage specificity to K48-specific ubiquitination, we assessed the impact of YOD1 on K48 polyubiquitination of RIPK2 and found that YOD1 deficiency increased the K48 polyubiquitination of RIPK2 (Fig. 6D). Furthermore, western blot analysis of immunocomplexes harvested from denatured lysates further confirmed that YOD1 deletion increased K48, rather than K63, ubiquitination directly linked to RIPK2 (Fig. 6E,F). Besides, YOD1 deficiency increased the colocalization of RIPK2 with the PSMD7$^+$ 26S proteasome (Fig. 6G), indicating that YOD1 can remove K48 polyubiquitin chains from RIPK2 and thereby inhibit its proteasomal degradation. Consistently, the reduced RIPK2 protein abundance in *Yod1$^{-/-}$* colons is accompanied with increased K48 ubiquitination of RIPK2 (Appendix Fig. S11A,B).

Structurally, YOD1 is a 343-aa protein containing a UBX-like (UBXL) domain (45–123 aa), an OTU domain (144–269 aa), and a C2H2 type zinc finger (ZnF) domain (313–337 aa). To pinpoint the domain that is required for binding RIPK2, we generated plasmids overexpressing truncated YOD1 (Fig. 6H). The co-IP result showed that the YOD1 mutant lacking the UBXL domain failed to interact with RIPK2, showing that YOD1 binds RIPK2 through the UBXL domain (Fig. 6I). The C155 and H262 residues located in the OTU domain are important for the deubiquitinating activity of YOD1. To pinpoint the activity site in YOD1 that mediates K48 deubiquitination of RIPK2, we constructed the YOD1 C155A mutant, whose C155 residue was inactive, and the YOD1 H262A mutant, whose H262 residue was inactive (Fig. 6J). As compared with full-length YOD1 and the H262A mutant, the C155A mutant could not increase the protein abundance of RIPK2, indicating that the C155 residue is required for YOD1 to K48 deubiquitinate and stabilize RIPK2 (Fig. 6K).

### Ablation of YOD1 abolishes the protective role of NOD2 signaling in DSS colitis

Since NOD2 signaling is protective in IBD, MDP administration protects mice from DSS-induced colitis and this protective effect of MDP is lost in mice with defective NOD2 signaling (Bertrand et al, 2009; Watanabe et al, 2008). To address the significance of YOD1 in NOD2 signal transduction in vivo, we studied the effect of MDP treatment on DSS colitis in *Yod1$^{+/+}$* and *Yod1$^{-/-}$* mice (Appendix

Fig. S15). In line with previous reports, administration of MDP significantly reduced body weight loss in *Yod1$^{+/+}$* mice upon DSS treatment (Fig. 7A). However, MDP administration did not influence the body weight loss of *Yod1$^{-/-}$* mice (Fig. 7A). Besides, MDP administration mitigated rectal bleeding, diarrhea, and colonic shortening in *Yod1$^{+/+}$* mice, but these protective effects were absent in *Yod1$^{-/-}$* mice (Fig. 7B–E). Consistent with the findings in Fig. 7A–E, the histopathological examination also found that MDP treatment had no impact on colonic pathology in *Yod1$^{-/-}$* mice (Fig. 7F–H; Appendix Fig. S16). In contrast to MDP administration, LPS treatment ameliorated colitis in both *Yod1$^{+/+}$* and *Yod1$^{-/-}$* mice (Fig. EV1A–I). However, colitis in *Yod1$^{-/-}$* mice was still significantly more severe than that in *Yod1$^{+/+}$* mice after LPS administration (Fig. EV1A–I), showing that YOD1 is dispensable for TLR4 signaling in vivo.

To directly confirm that YOD1 regulates colitis through RIPK2, we silenced RIPK2 in *Yod1$^{+/+}$* and *Yod1$^{-/-}$* mice with adeno-associated virus (AAV) (Fig. 8A,B). Consistent with the protective role of NOD2 signaling in colitis, deficiency of RIPK2 significantly exacerbated colitis in *Yod1$^{+/+}$* mice (Fig. 8C–J). Moreover, *Yod1$^{+/+}$* and *Yod1$^{-/-}$* mice had similar disease severity after RIPK2 knockdown, indicating that YOD1 affects colitis by regulating RIPK2. Taken together, these results demonstrate that YOD1 is indispensable for NOD2-induced protective effects in experimental colitis.

## Discussion

In the present study, we found that NOD2-mediated protective responses in colitis were potentiated by YOD1, which inhibited the proteasomal degradation of RIPK2 by removing K48-linked polyubiquitin chains. Ablation of YOD1 attenuated the activation of NOD2-mediated signal transduction and functional outcomes in macrophages, leading to exacerbated colitis. Therefore, this study demonstrates that YOD1 is an important protective regulator in colitis.

As an intestinal sentinel, NOD2 keeps luminal microbiota under close surveillance, safeguarding the intestinal homeostasis (Chu et al, 2016). NOD2 is an intracellular PRR that recognizes MDP, a PGN component conserved in both Gram-positive and Gram-negative bacteria (Girardin et al, 2003). Upon ligation with MDP, NOD2 oligomerizes and binds RIPK2. Subsequently, RIPK2 serves as a scaffold for the recruitment of TAK1, leading to the activation of downstream NF-κB and MAPK signaling (Ruan et al, 2022). Of note, ubiquitination plays an indispensable role in the activation and function of RIPK2, and TAK1 is recruited to polyubiquitinated RIPK2 via the adaptors TAB2 and TAB3 (Kanayama et al, 2004; Ruan et al, 2022). Accumulative studies have found that the activity of RIPK2 is vitally dependent on the K63-specific polyubiquitination, which is catalyzed by several E3s (TRAF6, XIAP, ITCH,

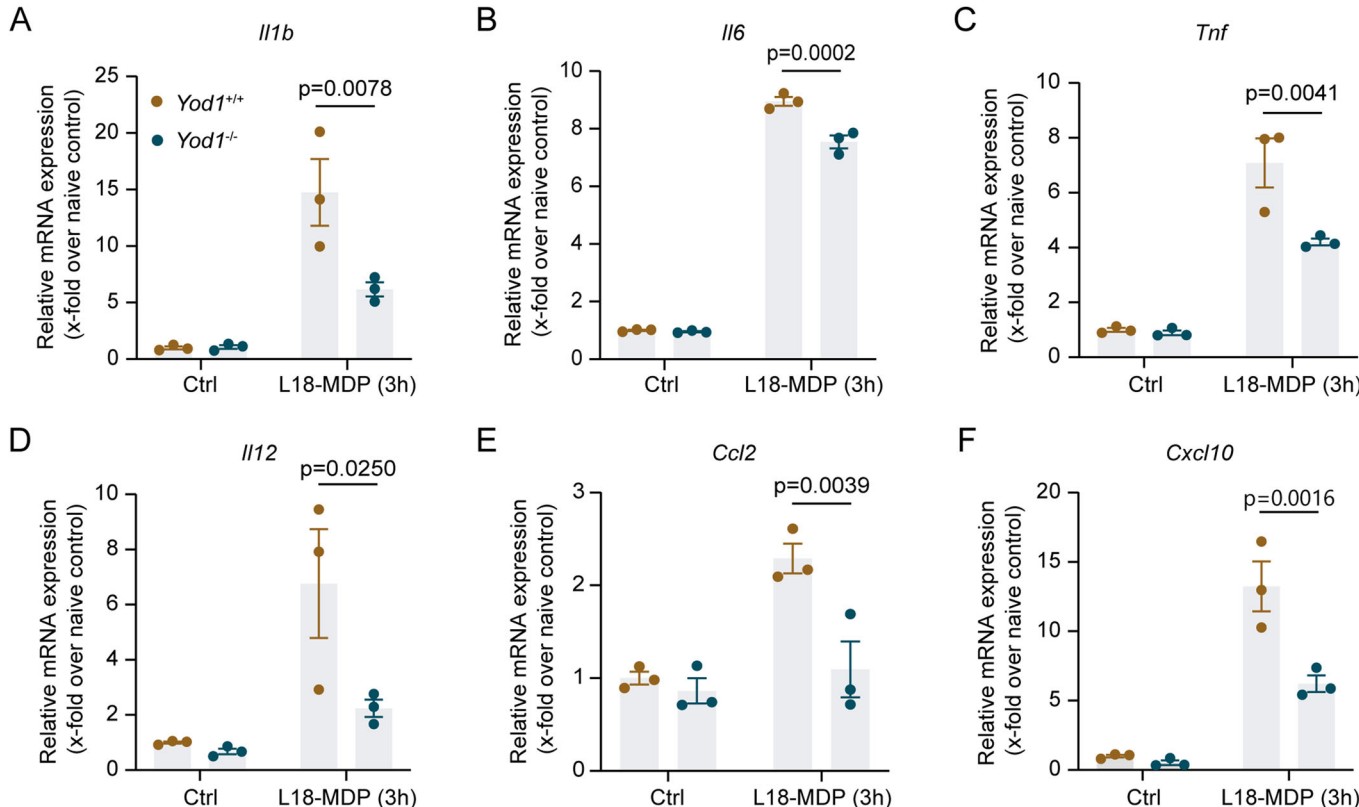

**Figure 4. YOD1-deficient BMDMs produce less cytokines and chemokines upon L18-MDP stimulation.**

(A–F) BMDMs isolated from *Yod1*^+/+ and *Yod1*^−/− mice were stimulated with L18-MDP (200 ng/ml) for 3 h or left untreated. The relative expression of *Il1b* (A), *Il6* (B), *Tnf* (C), *Il12* (D), *Ccl2* (E), and *Cxcl10* (F) mRNA was determined by qRT-PCR ($n = 3$, technical replicates). Data information: Data are representative of three replicates. Data show the mean ± SEM. Statistical analyses were performed using two-tailed unpaired Student's *t* test (A–F). Source data are available online for this figure.

RNF186, and Pellino3) and inhibited by some DUBs (CYLD, MYSM1, and A20) (Bertrand et al, 2009; Hitotsumatsu et al, 2008; Panda and Gekara, 2018; Ranjan Hedl and Abraham, 2021; Stafford et al, 2018; Tao et al, 2009; Yang et al, 2013). In addition to K63 ubiquitination, RIPK2 activity also requires linear ubiquitination, which is jointly catalyzed by XIAP and LUBAC (Damgaard et al, 2012). In sharp contrast to K63 and linear ubiquitination, K48 ubiquitination, which is mediated by the E3 ligase ZNRF4, leads to the degradation of RIPK2 (Bist et al, 2017). In this study, we found that the K48-specific polyubiquitination of RIPK2 was inhibited by the DUB YOD1. As far as we know, this is the first study showing that RIPK2 stability is enhanced by a DUB. Similar to NOD2, TLR4 is also a PRR that is applied by innate immune cells to recognize and respond to microbes. A previous study has shown that YOD1 inhibits IL-1-induced inflammatory responses (Schimmack et al, 2017), implying a potential role of YOD1 in TLR4 signaling. However, we found that YOD1 deficiency had no impact on LPS-induced signal transduction and cytokine production. The discrepancy warrants further studies, which may provide more convincing insights into the role of YOD1 in TLR4 signaling.

In line with published findings that NOD2^−/− and RIPK2^−/− mice are more susceptible to DSS-induced colitis than control mice (Couturier-Maillard et al, 2013), we found that *Yod1*^−/− mice, accompanied with reduced RIPK2 levels, also developed more severe experimental colitis than did control mice. Noteworthily,

YOD1 was mainly expressed in IECs under physiological conditions, but deletion of YOD1 had no impact on TNF-α-induced apoptosis of IECs, implying that YOD1 in IECs may be dispensable for colitis. During colitis, the expression of YOD1 was impaired in damaged IECs. Due to the overwhelming amounts of IECs, the overall expression of YOD1 was markedly reduced in the colon during colitis, indicating that the reduced colonic expression of YOD1 is the consequence, rather than the cause, of colitis. The bone marrow transplantation experiment unambiguously confirmed that the aggravated colitis in *Yod1*^−/− mice was predominantly caused by YOD1 deficiency in hematopoietic cells. In addition, we observed that YOD1 was upregulated in colon-infiltrating macrophages in both humans and mice during colitis, which may serve as a beneficial stress response to enhance the ability of macrophages to clear dissipated bacteria.

YOD1 is an enzyme comprising an N-terminal UBXL domain, a central OTU domain, and a C-terminal ZnF domain. The UBXL and ZnF domains are common binding sites for substrates, whereas the OTU domain exerts the DUB activity. The UBXL domain of YOD1 is essential for binding TRAF6 and p97, and YOD1 interacts with CDK1 via the C-terminal ZnF domain (Ernst et al, 2009; Han et al, 2023; Schimmack et al, 2017). Another study found that both the UBXL and ZnF domains were required for the interaction with MAVS (Liu et al, 2019). In this study, we demonstrate that YOD1 interacts with RIPK2 through the UBXL domain. The Cysteine

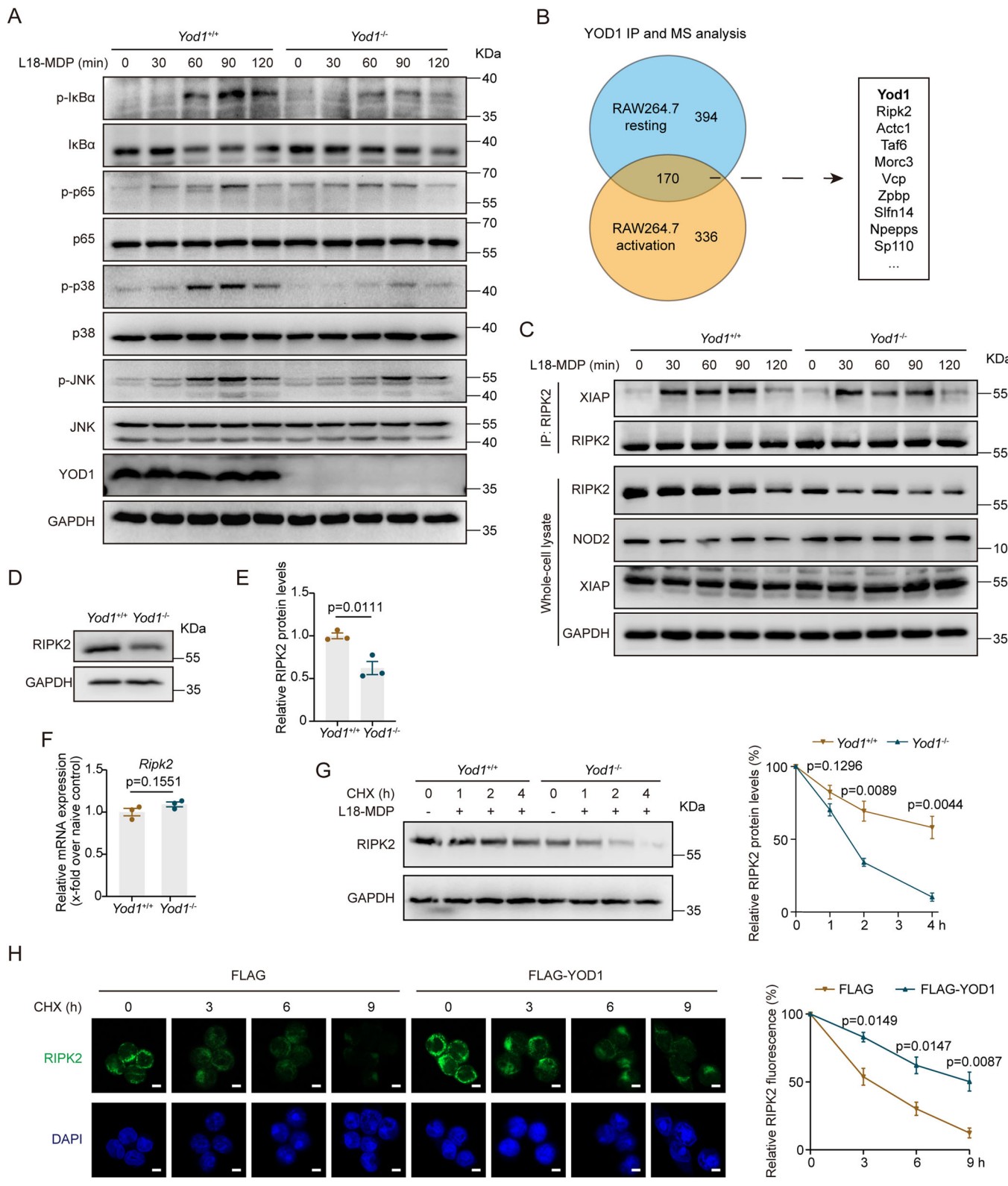

◀ **Figure 5. YOD1 enhances L18-MDP-induced signal transduction by increasing the abundance of RIPK2.**

(A) BMDMs isolated from $Yod1^{+/+}$ and $Yod1^{-/-}$ mice were stimulated with L18-MDP (200 ng/ml) for the indicated time points. Whole-cell lysates were analyzed by western blot with indicated antibodies. (B) RAW264.7 cells were left untreated (resting) or stimulated with LPS (500 ng/ml) for 30 min (activation). YOD1 and interacting proteins were immunoprecipitated from these cells and analyzed by mass spectrometry. (C) BMDMs isolated from $Yod1^{+/+}$ and $Yod1^{-/-}$ mice were stimulated with L18-MDP (200 ng/ml) for the indicated time points. Proteins immunoprecipitated with anti-RIPK2 antibody and whole-cell lysates were analyzed by western blot with indicated antibodies. (D, E) Representative western blot result (D) and relative quantification (E) of RIPK2 protein levels in BMDMs from $Yod1^{+/+}$ and $Yod1^{-/-}$ mice ($n = 3$, biological replicates). Mean ± SEM. *$P < 0.05$. (F) Relative levels of $Ripk2$ mRNA in BMDMs from $Yod1^{+/+}$ and $Yod1^{-/-}$ mice ($n = 3$, biological replicates). Mean ± SEM. ns, not significant. (G) Representative immunoblots (left) and quantification (right) of RIPK2 in BMDMs treated with CHX (20 ng/ml) and L18-MDP (200 ng/ml) for the indicated time points ($n = 3$, biological replicates). (H) RAW264.7 cells were transfected with FLAG or FLAG-YOD1 plasmids for 24 h, followed by treatment with CHX (20 ng/ml) for the indicated time points. Cells were then analyzed by immunofluorescence staining with anti-RIPK2 antibody (left). The right panel shows the relative immunofluorescence intensity of RIPK2 ($n = 3$, biological replicates). Scale bar = 5 μm. Data information: Data in (A, C–H) are representative of three replicates. The experiment in (B) was performed only once. Data in (E–H) show the mean ± SEM. Statistical analyses were performed using two-tailed unpaired Student's $t$ test (E–H). Source data are available online for this figure.

residue (C155 in mice and C160 in humans) in the OTU domain is critical for the DUB activity of YOD1 (Ernst et al, 2009; Kim et al, 2017; Wu et al, 2023). In contrast to the non-catalytic regulation of TRAF6 by YOD1 (Schimmack et al, 2017), we found that the YOD1 C155A mutant failed to stabilize RIPK2, showing that the DUB activity is required for the effect of YOD1 on RIPK2.

In summary, this study shows that YOD1 mediates optimal NOD2 signal transduction in macrophages and thereby ameliorates experimental colitis. To our knowledge, YOD1 is the first enzyme known to stabilize RIPK2, providing a novel and important mechanism in the regulation of NOD2 signaling. In addition, our data extend the in vivo function of YOD1 and broaden the spectrum of regulatory proteins in colitis. This study identifies YOD1 as a protective protein in colitis, indicating that therapeutic approaches enhancing YOD1 abundance or activity may be beneficial for IBD treatment.

# Methods

## Mice

C57BL/6 mice were purchased from Cyagen Biosciences (Suzhou, China). $Yod1^{-/-}$ mice were generated by Shanghai Model Organisms (Shanghai, China), and the flowchart for the generation of $Yod1^{-/-}$ mice is shown in Appendix Fig. S2A. Heterozygous $Yod1^{+/-}$ mice were bred to produce $Yod1^{+/+}$ and $Yod1^{-/-}$ littermates, and genotyping was carried out by PCR (forward: ACCAATTTTTCGTTTTCCCTGTGT; reverse: CTCCACAAGG CTTTCCACATTAC). All mice were kept in a specific-pathogen-free environment with 12 h of light and free access to food and water in the Laboratory Animal Resources Center of Wenzhou Medical University. Animal care and experiments were approved by the Animal Management and Ethics Committee of Wenzhou Medical University (Approval number: wydw2023-0154).

## Cell culture

RAW264.7 cells (Cat#: SCSP-5036), NIH/3T3 cells (Cat#: SCSP-515), and L-929 cells (Cat#: GNM28) were purchased from the National Collection of Authenticated Cell Culture (Shanghai, China). MODE-K cells (Cat#: BFN608006456) were purchased from Bluefbio (Shanghai, China). RAW264.7, NIH/3T3, and MODE-K cells were cultured in DMEM (Cat#: C11995500BT, Thermo Fisher Scientific, MA, USA) supplemented with 10% fetal bovine serum (FBS; Cat#: FSP500, ExcelBio, Taicang, China) and 1% penicillin/streptomycin Solution (Cat#: C0222, Beyotime

Biotechnology, Shanghai). L-929 cells were cultured in MEM-α medium (Cat#: C12571500BT, Thermo Fisher Scientific) with 10% FBS and 1% penicillin/streptomycin Solution.

## DSS-induced colitis model

Male mice at 8–12 weeks of age were treated with 2.5% DSS (MW 36,000–50,000; Cat#: 0216011080, MP Biomedicals) in drinking water for 5 days, followed by regular drinking water. Mice were co-housed for one week before DSS treatment to ensure that they have similar microbiota. For muramyldipeptide (MDP) treatment, each mouse was intraperitoneally administered with 100 μg of MDP (Cat #: A9519, Sigma-Aldrich) or volume-matched PBS for 3 consecutive days starting on the day of DSS challenge. For RIPK2 knockdown, mice were injected with AAV9-CON or AAV9-$Ripk2$-shRNA virus (Genechem, Shanghai, China) through the tail vein. Body weight, rectal bleeding, and stool consistency were monitored daily. Rectal bleeding was scored as follows: 0, absence of blood; 1, weak positive on the fecal occult blood test (Cat#: SK1030K-100T, G-clone, Beijing, China); 2, strong positive on the fecal occult blood test; 3, visible blood traces in stool; 4, gross rectal bleeding. Stool consistency was scored as follows: 0, normal; 1, soft but formed; 2, soft and loose; 3, very soft and wet; 4, watery diarrhea. On day 7 or 8, mice were sacrificed under anesthesia and the colons were isolated for the measurement of colon length, cytokine transcription, and histopathology. The spleen weight index was calculated as the spleen weight (mg) /the body weight (g).

## Plasmids construction and transfection

FLAG-YOD1, FLAG-YOD1-ΔUBX, FLAG-YOD1-ΔOTU, FLAG-YOD1-ΔZnf, Flag-YOD1-C155A, FLAG-YOD1-H262A, and the corresponding empty vector plasmids were purchased from Tsingke Biotechnology (Beijing, China). GFP-RIPK2 plasmids were purchased from Sino Biological (Beijing, China). All constructs were validated by DNA sequencing. Plasmids were transfected into RAW264.7 or NIH/3T3 cells using Lipofectamine 3000 reagent (Cat#: L3000015, Thermo Fisher Scientific) according to the manufacturer's instructions. Twenty-four hours after transfection, cells were used for further analysis.

## Bone marrow-derived macrophages (BMDMs)

The femur and tibia were aseptically isolated from mice at 6–8 weeks of age. Bone marrow cells were flushed out with ice-cold RPMI 1640 medium (Cat#: 11875093, Thermo Fisher

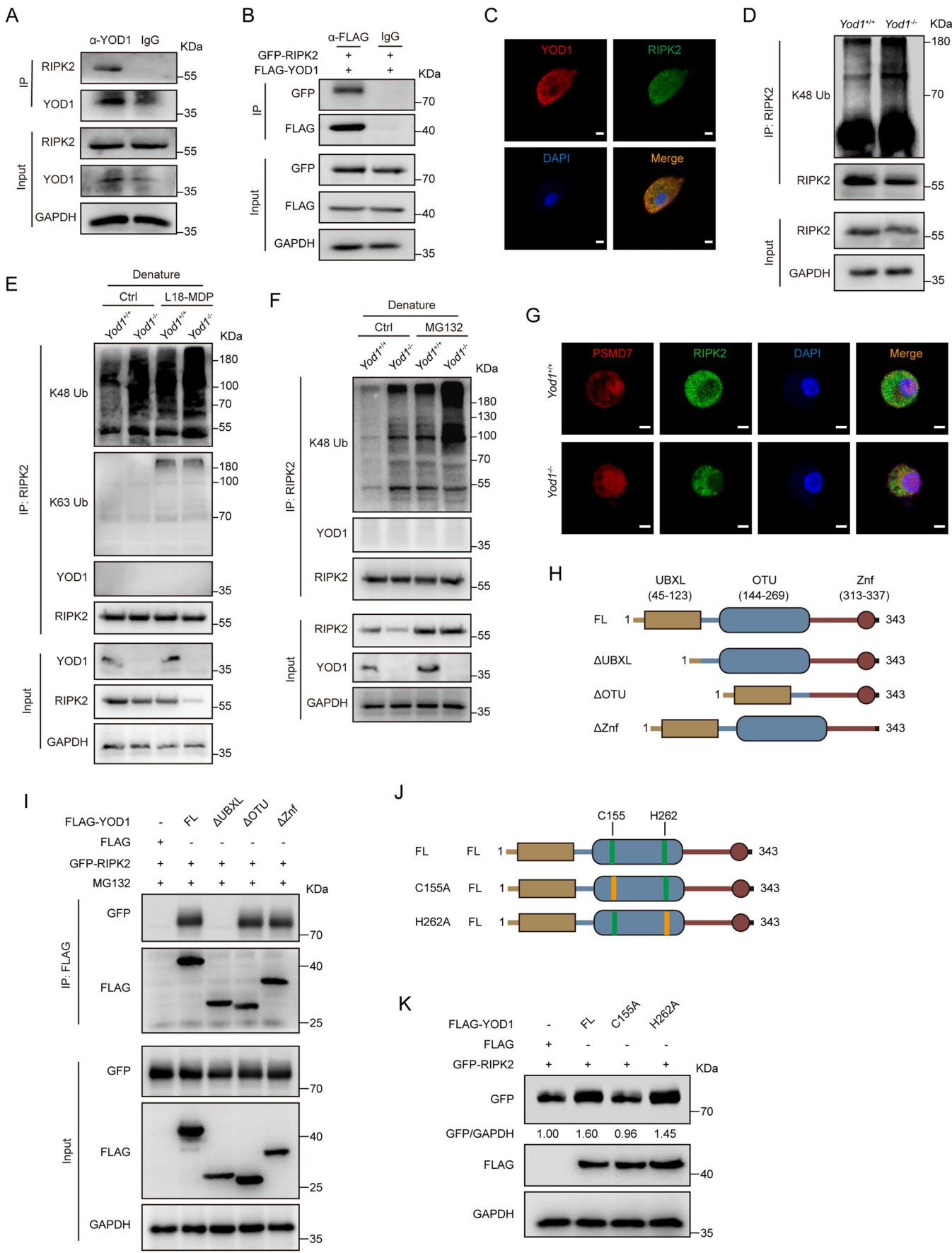

**Figure 6. YOD1 physically interacts with RIPK2 and reduces its K48 polyubiquitination.**

(A) Western blot analysis of proteins immunoprecipitated with anti-YOD1 antibody from BMDMs. (B) NIH/3T3 cells were transfected with GFP-RIPK2 and FLAG-YOD1 plasmids for 24 h. Proteins immunoprecipitated with anti-FLAG antibody were analyzed by western blot with indicated antibodies. (C) Immunofluorescence staining of YOD1 (red) and RIPK2 (green) in BMDMs. Original magnification, ×400. Scale bar = 5 μm. (D) BMDMs from *Yod1*$^{+/+}$ and *Yod1*$^{-/-}$ mice were treated with MG132 (20 μM) for 6 h before lysis. Proteins immunoprecipitated with anti-RIPK2 antibody were analyzed by western blot with indicated antibodies. (E, F) BMDMs from *Yod1*$^{+/+}$ and *Yod1*$^{-/-}$ mice were with L18-MDP (E) or MG132 (F) for 6 h before lysis. Proteins immunoprecipitated with anti-RIPK2 antibody from denatured cell lysates were analyzed by western blot with indicated antibodies. (G) Immunofluorescence staining of PSMD7 (red) and RIPK2 (green) in BMDMs from *Yod1*$^{+/+}$ and *Yod1*$^{-/-}$ mice. Original magnification, ×400. Scale bar = 5 μm. (H) Domain mapping of full-length (FL) YOD1 and truncated mutants. (I) NIH/3T3 cells were transfected with indicated plasmids for 24 h, and then treated with MG132 (20 μM) for 6 h before lysis. Proteins immunoprecipitated with anti-FLAG antibody were analyzed by western blot with indicated antibodies. (J) Schematic diagram of YOD1 mutants. (K) NIH/3T3 cells were transfected with indicated plasmids for 24 h. Whole-cell lysates were analyzed by western blot with indicated antibodies. Data information: Data in (A–D, G, I, K) are representative of three replicates. Data in (E, F) are representative of two replicates. Source data are available online for this figure.

Scientific) containing 1% penicillin/streptomycin and then filtered with 70-μm cell strainers. After lysing erythrocytes, cells were washed with PBS and then cultured in DMEM containing 20% L-929 culture medium and 10% FBS for 7 days. On days 3 and 5, cultures were replenished with fresh DMEM containing 20% L-929 culture medium and 10% FBS. BMDMs were stimulated with 200 ng/ml L18-MDP (Cat#: tlrl-lmdp, InvivoGen, CA, USA), 500 ng/ml LPS (Cat#: L2880, Sigma-Aldrich, MO, USA), 40 ng/ml TNF-α (Cat#: 315-01A, Peprotech), and PMA (Cat#: 50601ES03, YEASEN) for the indicated time points before further analysis.

## Bone marrow transplantation

The recipient mice were exposed to irradiation at a dose of 7.5 Gy. One week before and after irradiation, the recipient mice were given sterile drinking water containing 10 mg/ml neomycin sulfate (Cat#: HY-B0470, MedChemExpress, NJ, USA). Bone marrow cells were isolated from *Yod1*$^{+/+}$ and *Yod1*$^{-/-}$ mice, and $4.0 \times 10^6$ bone marrow cells were then intravenously injected into each recipient mouse through the tail vein. Eight weeks after transplantation, the recipient mice were challenged with 2.5% DSS to induce colitis. Splenocytes were isolated and then analyzed by western blot to verify the result of transplantation.

## Western blot

Animal tissues and cells were lysed in RIPA lysis buffer (Cat#: AR0105, Boster Bio, CA, USA) supplemented with protease and phosphatase inhibitor cocktail (Cat#: P1050, Beyotime Biotechnology) and PMSF (Cat#: P0100, Solarbio, Beijing, China). After centrifugation at 12,000× *g* at 4 °C for 10 min, the supernatant was collected and protein concentrations were determined with the Quick Start™ Bradford Protein Assay Kit 3 (Cat#: 5000203, Bio-Rad). After denaturing in SDS-PAGE Loading buffer (Cat#: FD006, FDbio, Hangzhou, China) at 100 °C for 5 min, protein samples were separated by SDS-PAGE and subsequently transferred to PVDF blotting membranes (Cat#: 10600023, Cytiva, Shanghai, China). Membranes were blocked with the NcmBlot blocking buffer (Cat#: P30500, NCM Biotech, Suzhou, China). Then, membranes were incubated with antibodies against YOD1 (Cat#: 25370-1-AP, Proteintech), NOD2 (Cat#: DF12125, Affinity Bioscience), XIAP (Cat#: 10037-1-Ig, Proteintech), p-IκBα (Cat#: 9246S, Cell Signaling Technology), IκBα (Cat#: 4814S, Cell Signaling Technology), p-p38 (Cat#: 4631S, Cell Signaling Technology), p38 (Cat#: 8690S, Cell Signaling Technology), p-p65 (Cat#: 3033S, Cell Signaling Technology), p65 (Cat#: 8242S, Cell Signaling Technology), p-JNK

(Cat#: 4668S, Cell Signaling Technology), JNK (Cat#: 9252S, Cell Signaling Technology), p-RIPK2 (Cat#: 14397, Cell Signaling Technology), RIPK2 (Cat#: 4142S, Cell Signaling Technology), K48 polyubiquitin (Cat#: 8081S, Cell Signaling Technology), FLAG (Cat#: 14793S, Cell Signaling Technology), GFP (Cat#: 50430-2AP, Proteintech), and GAPDH (Cat#:60004-1-Ig, Proteintech) at 4 °C overnight, followed by incubation with corresponding secondary antibodies at room temperature for 2 h. Thereafter, blots were developed with an Enhanced Chemiluminescent Kit (Cat#: P2300, NCM Biotech). Images were captured by the Fusion FX.EDGE system (Vilber, France) and analyzed by the ImageJ software (version 1.38e, NIH, Bethesda, USA).

## Immunoprecipitation

Protein samples were prepared as described in "Western blot". Equal amounts of protein samples were incubated with indicated antibodies (1.5 μg/mg protein) or IgG (Cat#: 30000-0-AP, Proteintech) under gentle rotation at 4 °C overnight. Thereafter, BeyoMag™ Protein A + G magnetic beads (Cat#: P2108, Beyotime) were added to the samples and incubated under gentle rotation at 4 °C for 2 h. The immunocomplexes were harvested by magnetic separation and subsequently washed with lysis buffer for five times before further analysis. For ubiquitination analysis, samples were denatured before immunoprecipitation as previously reported (Panda and Gekara, 2018; Panda Nilsson and Gekara, 2015). Briefly, lysates were suspended in 100 μl TSD buffer (50 mM Tris pH 7.5, 1% SDS, and 5 mM DTT) and boiled for 10 min. After centrifugation at 12,000 rpm for 5 min at room temperature, the supernatant was diluted with 1.2 ml TNN buffer (50 mM Tris pH 7.5, 250 mM NaCl, 5 mM EDTA, and 0.5% NP40). Corresponding antibodies were subsequently added for immunoprecipitation.

## Quantitative real-time PCR

Total RNA was isolated from animal tissues or cells with RNAiso plus (Cat#: 9109, Takara, Kyoto, Japan), and then reverse transcribed into cDNA with PrimeScript™ RT reagent Kit with gDNA Eraser (Cat#: RR047A, Takara). Quantitative Real-time PCR was performed with TB Green® Premix Ex Taq™ II (Cat#: RR820A, Takara) and corresponding primers on a QuantStudio™ 5 Real-Time PCR System (Thermo Fisher Scientific). All primers were synthesized by Sangon Biotech (Shanghai, China), and the sequences are shown in Appendix Table S1.

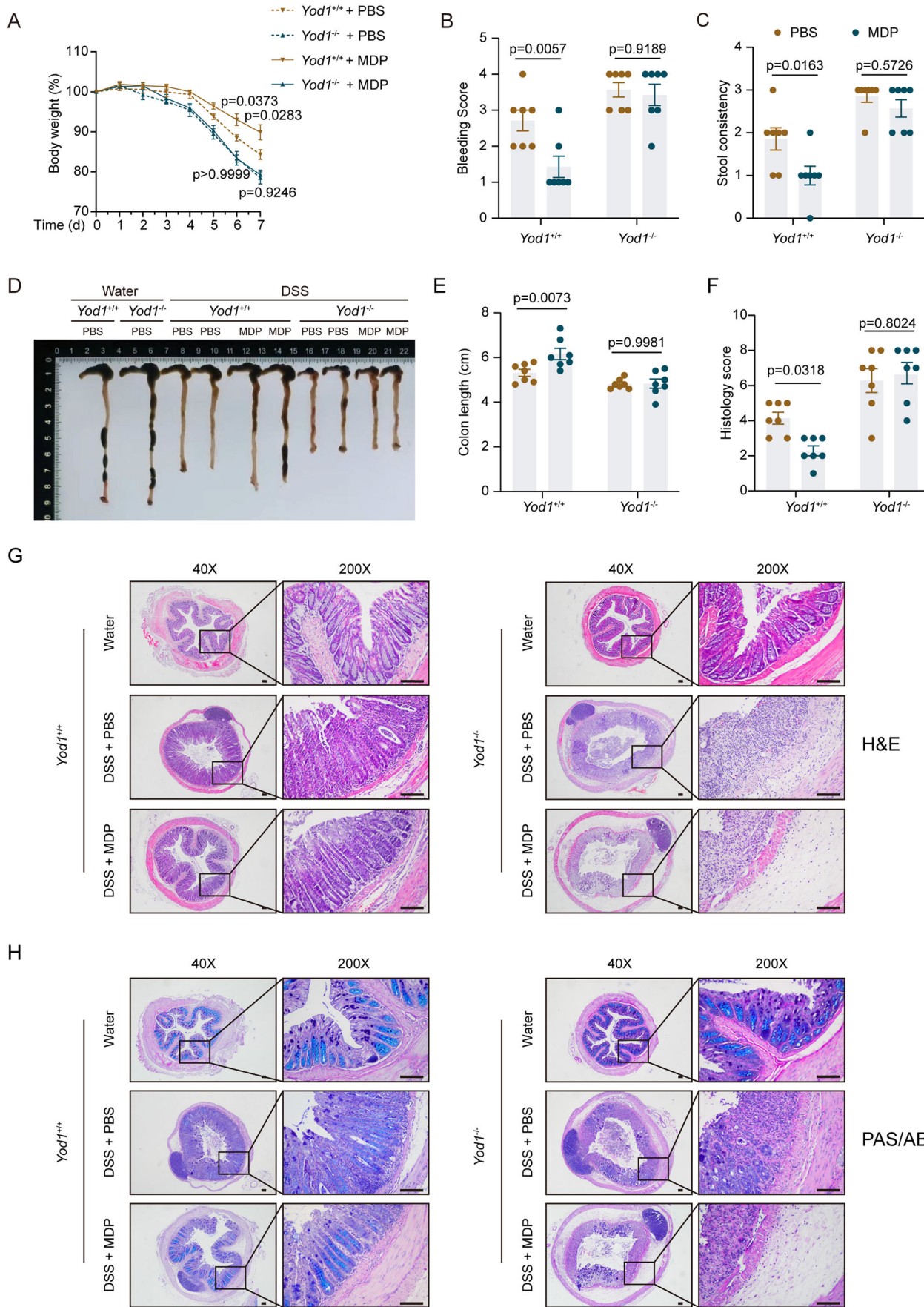

**Figure 7.  MDP treatment ameliorates DSS-induced colitis in *Yod1*<sup>+/+</sup> but not *Yod1*<sup>−/−</sup> mice.**

(A) Body weight change of *Yod1*[+/+] and *Yod1*[−/−] mice challenged with 2.5% DSS for 5 days followed by water for 2 days. Mice were intraperitoneally administered with MDP (100 μg/mouse) or PBS for 3 days starting on day 0 (*n* = 7, biological replicates). (B, C) Rectal bleeding (B) and stool consistency (C) of *Yod1*[+/+] and *Yod1*[−/−] mice on day 7 after DSS treatment (*n* = 7, biological replicates). (D–F) Representative image (D), length (E), and histology score (F) of colons from *Yod1*[+/+] and *Yod1*[−/−] mice on day 7 after DSS treatment (*n* = 7, biological replicates). (G, H) Representative H&E (G) and PAS/AB (H) staining of colons from *Yod1*[+/+] and *Yod1*[−/−] mice on day 7 after DSS treatment. Original magnification, ×40 and ×200. Scale bar = 100 μm. Data information: Data are representative of three replicates. Data in (A–C, E, F) show the mean ± SEM. Statistical analyses were performed using two-way ANOVA followed by the Sidak post-test (A–C, E, F). Source data are available online for this figure.

## Cycloheximide chase experiment

BMDMs or RAW264.7 cells were treated with 20 μg/ml cycloheximide (CHX; Cat#: 239763-M, Merck, Darmstadt, Germany) for the indicated times. Thereafter, cells were subjected to western blot or immunofluorescence analysis.

## Cell viability assay

MODE-K cells were inoculated in 96-well plates at a density of 5000 cells/well. For the induction of apoptosis, 40 ng/ml TNF-α (Cat#: 315-01A, Peprotech, NJ, USA) and 20 μg/ml CHX was added to cells for 6 or 12 h. Thereafter, 100 μl of DMEM complete medium containing 10% CCK8 solution (Cat#: 40203ES60, Yeasen, Shanghai, China) was added to each well and incubated at 37 °C for 1 h. The absorbance at 450 nm was measured on a Multiskan SkyHigh Microplate Spectrophotometer (Thermo Fisher Scientific).

## Clinical samples

Colon biopsy specimens were obtained from patients with UC at the First Affiliated Hospital of Wenzhou Medical University. In addition, tumor-adjacent normal colon tissues obtained from colon cancer patients who underwent colectomy surgery at the First Affiliated Hospital of Wenzhou Medical University were used as control colon tissues. Patient information is included in Appendix Table S2. The study on clinical samples was approved by the Ethics Committee in Clinical Research (ECCR) of the First Affiliated Hospital of Wenzhou Medical University (Approval number: KY2023-R182), and was performed in accordance with the Declaration of Helsinki.

## Histological analysis

Colon tissues were fixed with 4% paraformaldehyde (Cat#: P1110, Solarbio) for 48 h and then embedded with paraffin. Paraffin sections were cut at a thickness of 5 μm by a microtome, followed by dewaxing and hydration. subsequently, the sections were subjected to H&E staining (Cat#: G1120, Solarbio) to evaluate colonic damage and inflammation. Goblet cells were detected using periodic acid-Schiff/Alcian Blue (PAS/AB) staining (Cat#: G1285, Solarbio). Histological scores were displayed as the sum of the colonic damage score (0, normal morphology; 1, goblet cell loss; 2. large-scale loss of goblet cells; 3, crypt loss; 4, large-scale loss of crypts) and the inflammatory cell infiltration score (0, no infiltration; 1, infiltration with individual scattered inflammatory cells; 2, Infiltration spreading to the muscularis mucosa; 3, extensive infiltration of the muscularis mucosa with massive edema; 4, infiltration spreading to the submucosa).

## Enzyme-linked immunosorbent assay (ELISA)

BMDMs were seeded in the six-well plate at a density of $2 \times 10^6$ cells/well. Cells were treated with PMA (100 ng/ml), L18-MDP (200 ng/ml), TNF-α (40 ng/ml), and LPS (100 ng/ml) for 8 h. Thereafter, the supernatant was collected and measured with a commercially available ELISA kit (Cat#: E-EL-M0044, Elabscience, Wuhan, China) to detect IL-6 concentration.

## Flow cytometry

After washing away luminal contents, colons were incubated in HBSS (Cat#: H1040, Solarbio) containing 1 mM DTT, 1 mM EDTA, and 2% FBS under shaking at 220 rpm for 20 min at 37 °C. Subsequently, the tissue was sheared and digested in RPMI medium containing 2 mg/ml collagenase IV (Cat#: LS004186, Worthington Biochemical) at 220 rpm for 1 h at 37 °C. Macrophages (CD45[+] CD11b[+] F4/80[+]), B cells (CD45[+] CD19[+] CD45R[+]), and neutrophils (CD45[+] CD11b[+] Ly6G[+]) in the single cell suspension were analyzed by flow cytometry. Data were collected on a CytoFLEX LX Flow Cytometer (Beckman Coulter) and analyzed with the FlowJo software.

## Isolation of macrophages from colons

Single cell suspension was generated as described in "Flow cytometry". After cell counting, $3 \times 10^7$ cells were resuspended in 180 μl of PBS containing 0.5% BSA and 2 mM EDTA, followed by the addition of 30 μl of anti-F4/80 magnetic beads (Cat#: 92-01-0176, XinBio, Suzhou, China). After incubation for 20 min in the dark, cells were sorted on an autoMACS® NEO Separator (Miltenyi Biotec).

## Immunohistochemistry

Dewaxed and hydrated sections were boiled in 0.01 mol/L sodium citrate buffer for antigen retrieval, and then incubated with 3% $H_2O_2$ (Cat#: AR1108, Boster Bio) for 10 min at room temperature to inactivate endogenous peroxidases. After blocking with 5% BSA blocking buffer (Cat#: SW3015, Solarbio) for 30 min, the sections were incubated with antibodies against YOD1 (Cat#: 25370-1-AP, Proteintech) or F4/80 (Cat#: sc-377009, Santa Cruz Biotechnology) at 4 °C overnight. Horseradish peroxidase-conjugated secondary antibodies and 3,3′-diaminobenzidine (CAT#: ZLI-9018, ZSGB-BIO, Beijing, China) were used for signal detection. Subsequently, sections were dehydrated with an alcohol gradient and sealed with neutral balsam (Cat#: G8590, Solarbio). Images were captured with a Nikon ECLIPSE Ni-U microscope (Nikon, Tokyo, Japan) or a fully automated digital slide scanning system (Carl Zeiss AG, Oberkochen, Germany) and analyzed with ImageJ.

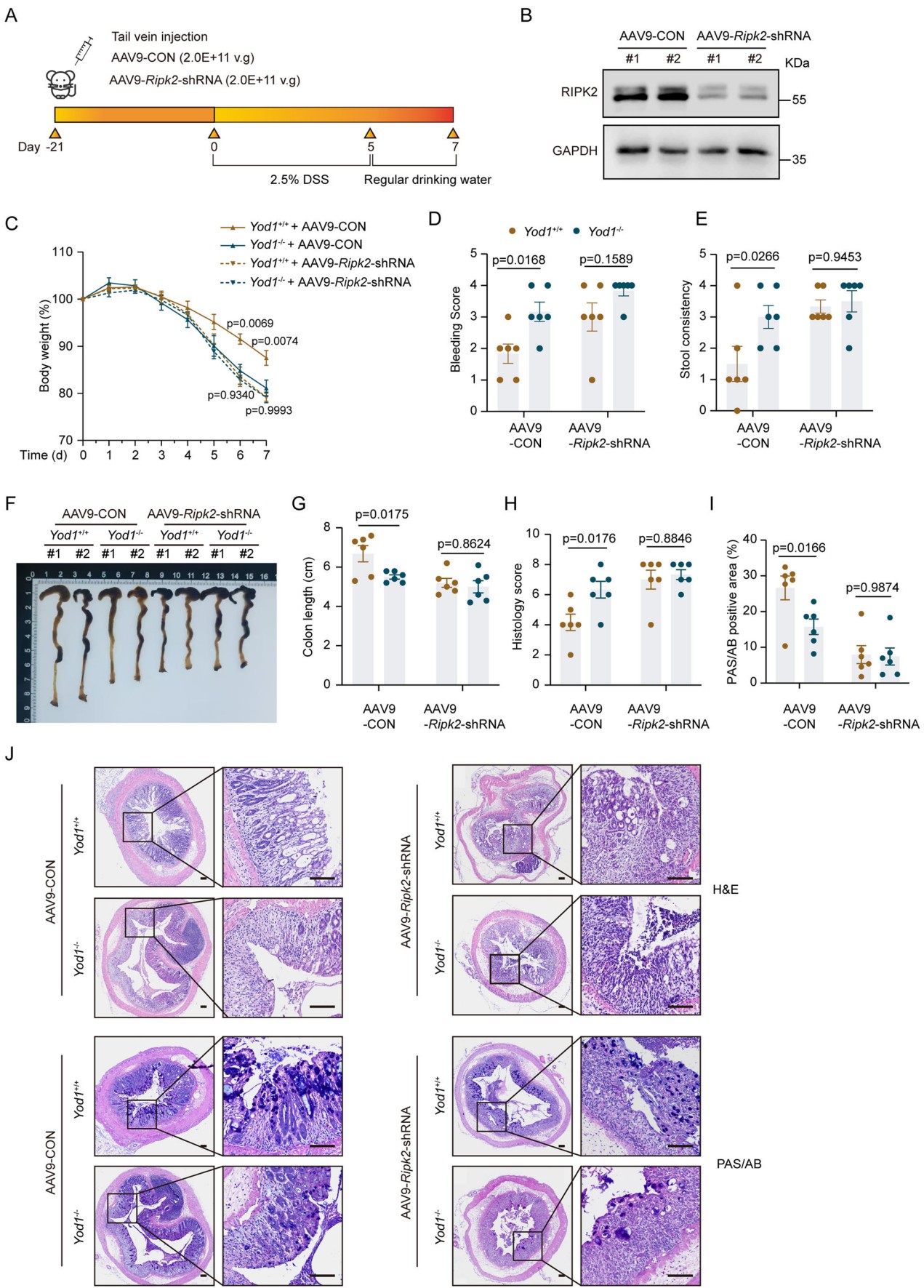

**Figure 8. RIPK2 silencing blunts the difference in colitis between $Yod1^{+/+}$ and $Yod1^{-/-}$ mice.**

(A) Experimental flowchart for AAV9 administration and DSS colitis. (B) RIPK2 expression in the spleen of AAV9-infected mice was analyzed by western blot. (C) Body weight change of AAV9-infected mice (n = 6, biological replicates). (D, E) Rectal bleeding (D) and stool consistency (E) of AAV9-infected mice on day 7 after DSS treatment (n = 6, biological replicates). (F–I) Representative image (F), length (G), histology score (H), and PAS/AB positive area (I) of colons from AAV9-infected mice on day 7 after DSS treatment (n = 6, biological replicates). (J) Representative H&E and PAS/AB staining of colons from AAV9-infected mice on day 7 after DSS treatment. Scale bar = 100 μm. Data information: Data are representative of two replicates. Data in (C–E, G–I) show the mean ± SEM. Statistical analyses were performed using two-way ANOVA followed by the Sidak post-test (C–E, G–I). Source data are available online for this figure.

## Immunofluorescence

Cells were fixed with 4% paraformaldehyde for 20 min and then permeabilized with 0.3% Triton X-100 (Cat#: T8200, Solarbio) for 20 min. Tissue sections were prepared as described in "Histological analysis" and "Immunohistochemistry". After blocking with 5% BSA blocking buffer for 1 h at room temperature, samples were incubated with primary antibodies against RIPK2 (Cat#: sc-136059, Santa Cruz Biotechnology), F4/80 (Cat#: sc-377009, Santa Cruz Biotechnology), YOD1 (Cat#: 25370-1-AP, Proteintech), PSMD7 (Cat#: 16034-1-AP, Proteintech) at 4 °C overnight, followed by incubation with secondary antibodies conjugated with Alexa Fluor 488 (Cat#: 33206ES60, Yeasen) or 594 (Cat#: 34212ES60, Yeasen) at room temperature for 2 h. Samples were counterstained with DAPI (Cat#: S2110, Solarbio). Images were captured on a ZEISS LSM 980 with Airyscan 2 confocal microscope (Carl Zeiss AG, Oberkochen, Germany).

## Statistical analyses

Statistical analyses were carried out using GraphPad Prism 8 (GraphPad, CA, USA). The two-tailed Student's t test was used to compare data between two groups. One-way or two-way ANOVA test was applied to compare more than two groups of data. P values < 0.05 were considered significant.

## Data availability

This study includes no data deposited in external repositories.

The source data of this paper are collected in the following database record: biostudies:S-SCDT-10_1038-S44319-024-00276-6.

## Peer review information

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

## Acknowledgements

This work was supported by grants from the Natural Science Foundation of Zhejiang Province (LZ24H090003) and the National Natural Science Foundation of China (81900496) to XW.

## Author contributions

**Jiangyun Shen**: Conceptualization; Resources; Data curation; Formal analysis; Investigation; Visualization; Methodology; Writing—original draft; Writing—review and editing. **Liyan Lou**: Data curation; Investigation; Visualization; Methodology; Writing—review and editing. **Xue Du**: Investigation; Methodology. **Bincheng Zhou**: Investigation; Methodology. **Yanqi Xu**: Investigation; Methodology. **Fuqi Mei**: Resources; Investigation. **Liangrong Wu**: Resources; Investigation. **Jianmin Li**: Resources; Writing—review and editing. **Ari Waisman**: Validation; Writing—review and editing. **Jing Ruan**: Conceptualization; Resources; Formal analysis; Validation; Project administration; Writing—review and editing. **Xu Wang**: Conceptualization; Supervision; Funding acquisition; Validation; Writing—original draft; Project administration; Writing—review and editing.

Source data underlying figure panels in this paper may have individual authorship assigned. Where available, figure panel/source data authorship is listed in the following database record: biostudies:S-SCDT-10_1038-S44319-024-00276-6.

## Disclosure and competing interests statement

The authors declare no competing interests.

# Expanded View Figure

**Figure EV1.  LPS treatment ameliorates DSS colitis in both *Yod1*$^{+/+}$ and *Yod1*$^{-/-}$ mice.**

(A) Experimental flowchart for the DSS experiment with LPS treatment. (B) Body weight change of *Yod1*$^{+/+}$ and *Yod1*$^{-/-}$ mice ($n = 6$, biological replicates). (C, D) Rectal bleeding (C) and stool consistency (D) of *Yod1*$^{+/+}$ and *Yod1*$^{-/-}$ mice on day 7 after DSS treatment ($n = 6$, biological replicates). (E–H) Representative image (E), length (F), histology score (G), and PAS/AB positive area (H) of colons from *Yod1*$^{+/+}$ and *Yod1*$^{-/-}$ mice on day 7 after DSS treatment ($n = 6$, biological replicates). (I) Representative H&E and PAS/AB staining of colons from *Yod1*$^{+/+}$ and *Yod1*$^{-/-}$ mice on day 7 after DSS treatment. Scale bar $= 100$ µm. Data information: Data are representative of two replicates. Data in (B–D, F–H) show the mean ± SEM. Statistical analyses were performed using two-way ANOVA followed by the Sidak post-test (B–D, F–H). Source data are available online for this figure.

   

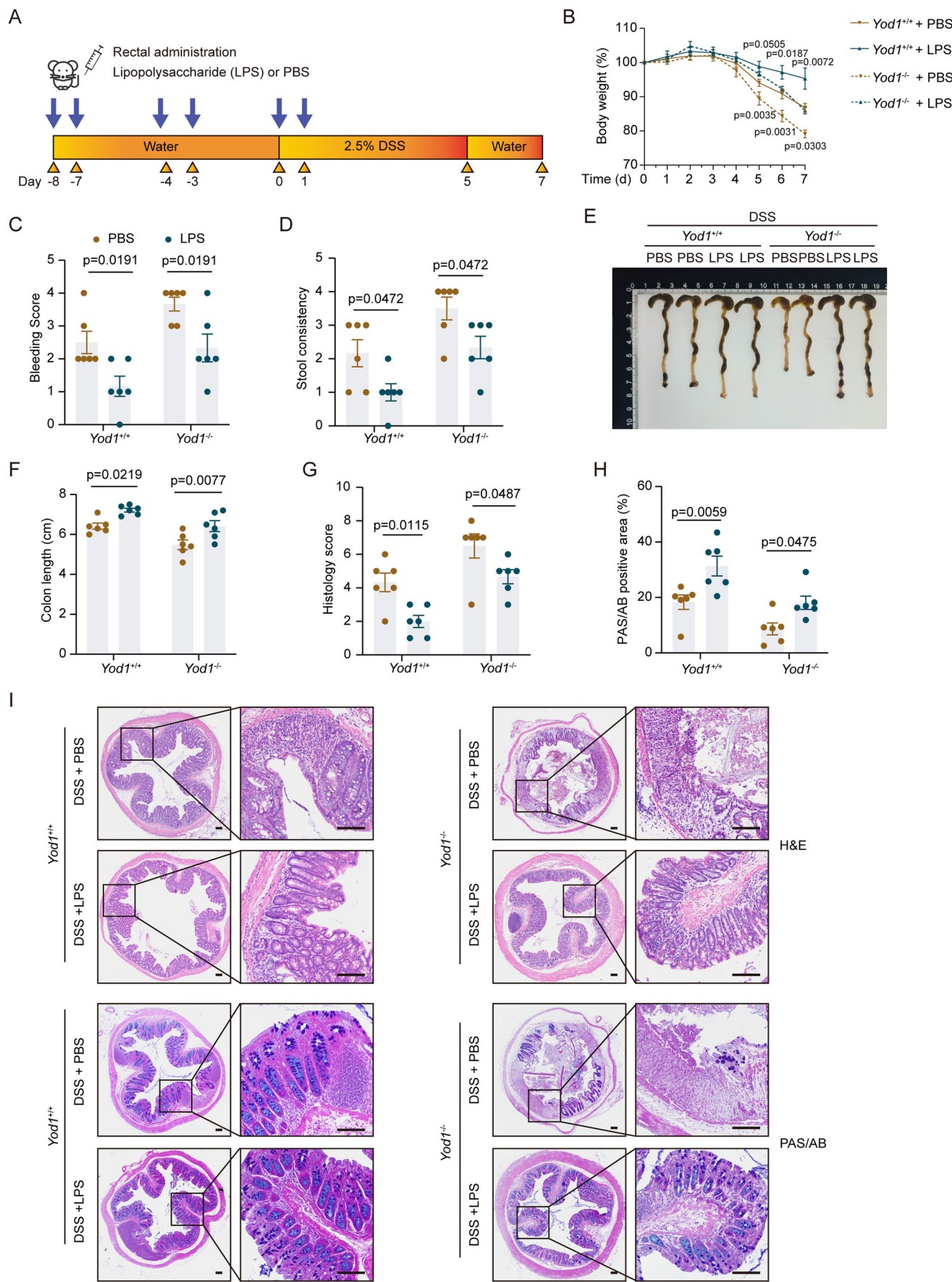

