## [Peer Review File · EMBO Reports]

YOD1 sustains NOD2-mediated protective signaling in colitis by stabilizing RIPK2

Jiangyun Shen, Liyan Lou, Xue Du, Bincheng Zhou, Yanqi Xu, Fuqi Mei, Liangrong Wu, Jianmin Li, Ari Waisman, Jing Ruan, and Xu Wang

Corresponding author(s): Jing Ruan (ruanjing@wzhospital.cn, ruanjing850617@163.com), Xu Wang (wangxu@ojlab.ac.cn, sunrim@163.com)

Review Timeline:

Submission Date:	8th Mar 24
Editorial Decision:	4th Apr 24
Revision Received:	31st Jul 24
Editorial Decision:	22nd Aug 24
Revision Received:	3rd Sep 24
Accepted:	18th Sep 24

Editor: Achim Breiling

Transaction Report:

Dear Prof. Wang,

Thank you for the submission of your research manuscript to EMBO reports. I have now received the reports from the three referees that were asked to evaluate your study, which can be found at the end of this email.

As you will see, the referees think that the findings are of interest. However, the referees have several comments, concerns, and suggestions, indicating that the data are rather preliminary and that a major revision of the manuscript is necessary to allow publication of the study in EMBO reports. From the analysis of the referee comments it is clear that a significant revision is required before publication can be considered, and I would also understand your decision if you chose to rather seek rapid publication elsewhere at this stage.

However, I would like to give you the opportunity to address the concerns and would be willing to consider a revised manuscript with the understanding that all referee concerns must be addressed in the revised manuscript and in a detailed point-by-point response. Acceptance of your manuscript will depend on a positive outcome of a second round of review. It is EMBO reports policy to allow a single round of revision only and acceptance of the manuscript will therefore depend on the completeness of your responses included in the next, final version of the manuscript.

- 1) a .docx formatted version of the final manuscript text (including legends for main figures, EV figures and tables), but without the figures included. Figure legends should be compiled at the end of the manuscript text.
- 2) individual production quality figure files as .eps, .tif, .jpg (one file per figure), of main figures and EV figures. Please upload these as separate, individual files upon re-submission.

- 4) a complete author checklist, which you can download from our author guidelines

(<https://www.embopress.org/page/journal/14693178/authorguide>). Please insert page numbers in the checklist to indicate where the requested information can be found in the manuscript. The completed author checklist will also be part of the RPF.

5) that primary datasets produced in this study (e.g. RNA-seq, CHIP-seq, structural and array data) are deposited in an appropriate public database. If no primary datasets have been deposited, please also state this in a dedicated section (e.g. 'No primary datasets have been generated and deposited'), see below.

The accession numbers and database should be listed in a formal "Data Availability" section (placed after Materials & Methods) that follows the model below. This is now mandatory (like the COI statement). Please note that the Data Availability Section is restricted to new primary data that are part of this study. This section is mandatory. As indicated above, if no primary datasets have been deposited, please state this in this section

Data availability

8) Regarding data quantification and statistics, please make sure that the number "n" for how many independent experiments were performed, their nature (biological versus technical replicates), the bars and error bars (e.g. SEM, SD) and the test used to calculate p-values is indicated in the respective figure legends (also for EV figures and all those in an Appendix). Please also check that all the p-values are explained in the legend, and that these fit to those shown in the figure. Please provide statistical testing where applicable. Please avoid the phrase 'independent experiment', but clearly state if these were biological or technical replicates. Please also indicate (e.g. with n.s.) if testing was performed, but the differences are not significant. In case n=2, please show the data as separate datapoints without error bars and statistics. See also: <http://www.embopress.org/page/journal/14693178/authorguide#statisticalanalysis>

9) Please add scale bars of similar style and thickness to microscopic images, using clearly visible black or white bars (depending on the background). Please place these in the lower right corner of the images themselves. Please do not write on or near the bars in the image but define the size in the respective figure legend.

10) Please also note our reference format:

12) We now use CRediT to specify the contributions of each author in the journal submission system. CRediT replaces the author contribution section. Please use the free text box to provide more detailed descriptions and do not provide your final manuscript text file with an author contributions section. See also our guide to authors: <https://www.embopress.org/page/journal/14693178/authorguide#authorshipguidelines>

13) We would encourage you to use 'Structured Methods', our new Materials and Methods format. According to this format, the Materials and Methods section should include a Reagents and Tools Table (listing key reagents, experimental models, software, and relevant equipment and including their sources and relevant identifiers), uploaded as separate file, followed by a Methods and Protocols section in which we encourage the authors to describe their methods using a step-by-step protocol format with bullet points, to facilitate the adoption of the methodologies across labs. More information on how to adhere to this format as well as downloadable templates (.doc or .xls) for the Reagents and Tools Table can be found in our author guidelines (section 'Structured Methods'):

14) Please order the manuscript sections like this, using these names:

Title page - Abstract - Keywords - Introduction - Results - Discussion - Methods - Data availability section - Acknowledgements - Disclosure and Competing Interests Statement - References - Figure legends - Expanded View Figure legends

Finally, please note that all corresponding authors are required to supply an ORCID ID for their name upon submission of a revised manuscript. Please do that for co-corresponding author Ruan. Please find instructions on how to link the ORCID ID to the account in our manuscript tracking system in our Author guidelines:

<http://www.embopress.org/page/journal/14693178/authorguide#authorshipguidelines>

I look forward to seeing a revised version of your manuscript when it is ready. Please let me know if you have questions or comments regarding the revision.

Yours sincerely,

Referee #1:

The manuscript by Shen et al. describes the activity of Yod1, a K48 and K63 deubiquitinase on the pathogenesis of a mouse model of Ulcerative colitis-like pathology. Yod1 has previously been shown to be involved in multiple innate immune signaling pathways, including IL-1 (Shimmack et al., eLife, 2017) and MAVS (Lui et al, J. Immunology, 2019) and importantly, shows a well-published role in ER dynamics (Ernst et al., Mol Cell, 2009; Bernardi et al., Mol.Bio.Cell, 2013; etc). In the current work, Shen et al. link the DSS-colitis effect to the K48 deubiquitination of the scaffolding protein Kinase RIPK2, a member of the innate immune NOD1/2 signaling system. They then claim that Yod1 regulates RIPK2 stability.

These findings would be of interest to the innate immune community, but as submitted, the authors really fail to link Yod1 and RIPK2 together. They don't rule out ER involvement nor do they rule out IL-1 or MAVS involvement. In fact, the data linking Yod1 to RIPK2 are unconvincing and if anything, to this reviewer, don't show an effect of Yod1 to RIPK2 stability. For these reasons, the work seems preliminary and somewhat overstated and for this reason, enthusiasm is muted. My specific comments are as follows:

1. The connection here between Yop1 and RIPK2 is tenuous. The signal defect with L18-MDP in Figure 5A is minimal at best. Figure 5c shows loss of phospho-274 RIPK2, but this has not been consistently seen as MDP-inducible. Most labs use RIPK2 ubiquitination and/or XIAP binding as more convincing evidence of RIPK2 activity. In fact in Figure 5C, there's no difference in the half-life of RIPK2 upon L18-MDP stimulation in Yop1-null cells and then amounts of RIPK2 present are nearly identical in the Yod1-WT and Yod1-null BMDMs and in Figure 5D, the amounts of RIPK2 present don't seem very different - as quantified in Figure 5D, its unlikely this small of a difference causes a biological effect. The half-life differences in Figure 5G needs to be performed +/- MG132 (see transfection level concerns below) and the graph in 5H isn't really convincingly different. In Figure 6D, the amount of K48 ubiquitination is not noticeably different in a biological sense. The only experiment really linking the NOD2/RIPK2 pathway with Yod1 is the MDP-DSS protection experiment in Figure 7, but MDP or any other innate immune stimulus - esp LPS - is known to protect against DSS colitis. RIPK2, itself, has given variable results in DSS colitis assays with most studies showing little or no effect of RIPK2-WT or RIPK2-null mice in DSS assays.

On top of the above, Yop1 has been shown to interact with an affect TRAF6 - a major player in TLR and IL-1 signaling and has been shown to effect ER dynamics. In my mind, its far more likely that Yop1 has a more general role in innate immune signaling - most likely at a cell biological level - than a specific effect on RIPK2, especially given the data shown in Figures 3 and 6. Lastly, RIPK2 ubiquitination is known to be regulated by other deubiquitinases including A20 and CYLD (Abbott et al., Current biology,

2004; Hitosumatsu et al., Immunity 2008). Its likely that there's redundancy in RIPK2 deubiquitination and for this reason, the singular effect of Yod1 needs far more proof than what is provided.

2. Continuing on point 1 above, but from a technical standpoint, there are a number of issues with the molecular biology in the manuscript. In Figure 4, RAW macrophages are notoriously difficult to transfect and can initiate an innate immune response to intracellular DNA when transfected. Good transfection efficiencies are around 25% (see Cheung et al JOVE, 2015 for a good summary) and there's considerable cellular response to DNA. Given that, it is not surprising that Figure 5G shows minimal results when FLAG-Yod1 is transfected. In these types of experiments, one would typically make an inducible Yod1 in a stably transduced viral system to overcome transfection efficiencies and effects. Additionally, in Figure 6, when looking at ubiquitinated proteins, one typically needs to denature the lysate before performing the IP. This is to rid the target protein of ubiquitinated binding partners - these can give artifactual findings. This is particularly important with RIPK2 as it forms a complex with proteins known to be ubiquitinated including XIAP and NEMO. Figure 6D needs to do this. Lastly, the ubiquitination experimentation is underdone. Yod1 appears to have K63 and K48 deubiquitinase activity. Does Yod1 deubiquitinate K63-linked RIPK2? What is K48-ubiquitinating RIPK2 and does Yod1 interact with that? Half life experiments would be better performed with MDP-stimulation - are the authors supposing that Yod1 regulates basal, unstimulated RIPK2? RIPK2 expression is known to be induced by inflammatory stimuli like LPS or TNF (Kobayashi et al., Nature, 2005). Does this induction affect Yod1 activity on RIPK2? Is Yod1 specific to RIPK2? Does it affect TRAF6 as well and could this underlie some of the results seen?

3. RAW264.7 cells are not the best in which to study a NOD2:RIPK2 signaling axis. People typically use PMA-differentiated THP-1 cells or immortalized BMDMs for biochemical experiments as the signaling is much stronger. This may be one reason the results in Figure 5 aren't robust.

4. Given all this, I find it far more likely that Yod1 affect innate immunity broadly. The authors should test LPS, PamCys, PMA, IL-1, etc on the BMDMs and see if there's a more broad innate immune effect. Figure 4, for instance, would benefit from comparison with other innate immune stimuli. "Relative expression" really doesn't give the reader a sense of levels or effect. These levels should be formally tested by ELISA. MDP likely gives a 100-fold lower effect than LPS.

5. Within the field, there hasn't really been a phenotype of RIPK2-null mice with DSS. Most experiments have needed to use TNBS, Samp/YitFc, TRUC mice on Balb/C background to see an effect. There's also the well-known role of the microbiota on DSS experiments and on NOD2 pathology in general. These don't seem to have been accounted for in the experimentation. At the very least, the authors should use co-housed mice with RIPK2 inhibitors and see if there's an additive effect on the Yod1-null mice. Additionally mating RIPK2-null mice with Yod1 mice could be performed to see if there's an additive effect.

6. Its well-known that innate immune priming (typically with LPS, but also with MDP) can protect against DSS colitis. A LPS-protective effect on DSS colitis should still be seen in the Yod1 mice.

7. Figure 7 needs to look at RIPK2 ubiquitination and RIPK2 levels in the mice. RIPK2 levels should be much lower in the Yod1-null mice.

In summary, I find the link between Yod1 and RIPK2 to be tenuous. Yod1 is more likely affecting the innate immune system more broadly - like through ER dynamics, but also through IL-1 and MAVS signaling (as published). RIPK2 may be playing a role here, but other innate immune pathways may predominate, especially as RIPK2's role in DSS colitis has been variable across labs. Given these concerns, coupled with the molecular biology issues outlined above, enthusiasm is muted.

Referee #2:

Shen et al. discover that patients with ulcerative colitis exhibit elevated levels of YOD1. Their study suggests that YOD1 plays a regulatory role in the NOD2 pathway. Through in vivo experiments involving bone marrow transplants, the authors convincingly demonstrate that immune cells lacking YOD1 exacerbate inflammation following DSS colitis. This finding underscores the protective role of YOD1. Additionally, the study highlights a reduction in YOD1 expression alongside an increase in F4/80 cell infiltration in both DSS-treated mice and ulcerative colitis patients. Consistent with existing literature, activating the NOD2 pathway via MDP treatment conveys protective effects in DSS-treated mice, this protection is significantly diminished by YOD1 deficiency. The authors employ a range of cell biology and biochemistry techniques to link YOD1 deficiency with decreased NOD2-dependent NF- κ B signalling and cytokine production. They attribute these outcomes to lowered RIPK2 levels, positing that this is due to heightened K48-ubiquitination of RIPK2 in the absence of YOD1's catalytic activity.

The in vivo data presented is of high quality, and the elucidated role of YOD1 in DSS-colitis will be of interest to those in the NOD2 research community, hinting at a novel therapeutic target for managing NOD2-driven ulcerative colitis. However, the paper should clarify whether YOD1's effect on NF- κ B signalling occurs via a TRAF6/TNF-dependent or independent pathway. While the mechanistic insights (from cell biology and biochemistry studies) mostly suggest correlation, further evidence is needed to confirm that YOD1 directly influences the NOD2 pathway, and that this influence is the driving force for the in vivo phenotype.

Crucial Concerns:

- Referencing Stafford et al. Cell Reports (2018), which demonstrates the interplay between NOD2 and the TNF pathway in cytokine response, the paper should address whether TRAF6 and TNFR1 might account for the observed effects, given YOD1's known regulatory impact on TRAF6. Additionally, given RIPK2's precise protein level regulation, any basal changes in NF- κ B signalling in YOD1 knockouts could influence RIPK2 levels, and therefore NOD2 signalling. Looking at Fig 5C, it appears that NOD2 levels might be increased in the KO, hinting towards such an event as NOD2 is also sensitive to NF- κ B changes.
- Although interactions between YOD1 and RIPK2 are noted, this does not rule out YOD1's potential impact on other pathways. The observed phenotypes and weak biochemical evidence regarding YOD1's influence on RIPK2's K48 ubiquitination call for more robust investigation.
- The authors reference Bist et al, 2017 and the K48 ubiquitination of RIPK2. In that study, they indicate RIPK2 is K48-ubiquitinated following MDP treatment, rather than basal RIPK2 regulation. In this study, there is a lack of investigation into whether YOD1 impacts basal or NOD2 activated RIPK2 levels, especially considering that a major finding of the study is MDP treatment protects DSS-colitis in an MDP dependent manner.

Suggested Experiments:

1. To verify that the NOD2 cytokine phenotype stems from YOD1's action on RIPK2 rather than the TNF pathway, compare cytokine responses in WT and YOD1 KO BMDMs and primary cells from the peritoneum of mice, with and without a TNF blocking antibody, upon NOD2 activation.
2. Investigate whether the TNF pathway significantly contributes to the observed differences in WT versus YOD1 BMDM responses to TNF treatment (TNF timecourse and Western blotting of NF- κ B/MAPK activation).
3. Strengthen the evidence that YOD1 directly affects K48 ubiquitination on RIPK2 through specific experiments designed to isolate this interaction in a time dependent manner (+/- MG132, +/- MDP treatment, Tandem Ubiquitin Binding Entities TUBEs, then blot for RIPK2 to ensure ubiquitination is directly linked to RIPK2).

Minor concerns:

- Improve and strengthen evidence and clarity in Figure 2g's depiction of F4/80 staining and macrophage infiltration in WT vs. YOD1 KO mice. Ensure arrows clearly point to the relevant cells and provide higher-quality images.
 - Clarify whether mass spectrometry results from pulldowns (hit list), will be published, and whether TRAF6 was shown to interact.
 - For further validation, include all ladder markers in the raw and published Western blots in both the review process and final manuscript.
 - Correct the discussion on the role of cIAP1/2 and XIAP in RIPK2 ubiquitination, referencing Stafford et al. (2018) to accurately represent XIAP's primary role over cIAP1/2 for K63 linked ubiquitination of RIPK2.
- Addressing these points will significantly enhance the manuscript's contribution to the field, offering a clearer and more compelling narrative of YOD1's role in regulating the NOD2 pathway in ulcerative colitis.

Referee #3:

The manuscript "YOD1 sustains NOD2-mediated protective effects in colitis by stabilizing RIPK2" by Shen and colleagues explores a hitherto unknown regulator of intestinal inflammation. The authors first describe an aberrant colonic expression pattern of the deubiquitinase, YOD1, in ulcerative colitis (UC) and during murine DSS-induced colonic inflammation. Then, the group generated Yod1-deficient mice and subjected them to an acute stimulus of DSS and compared their colonic pathologies relative to littermate wildtype (WT) mice. The group found that YOD1-deficiency resulted in a greater intestinal inflammatory burden in various readouts of intestinal inflammation. Using a sequence of gene transcription, bone marrow chimera, and immunostaining experiments the authors suggest that YOD1-deficiency specifically in macrophages drives this increased inflammation. Specifically, the authors suggest that YOD1-deficiency induces increased macrophage infiltration into the colons of DSS-treated mice, which is associated with an increase in colonic proinflammatory gene transcription. To explore the macrophage-intrinsic role of YOD1, the authors treated WT and YOD1 KO BMDMs with the synthetic NOD2-agonist, L18-MDP, and found that YOD1-deficiency lead to decreased proinflammatory gene transcription following NOD2 agonism. Mass-spectrometry and Co-IP experiments suggest that YOD1 physically interacts with RIPK2; a downstream regulator of NOD2-mediated signalling. More specifically, the authors suggest that YOD1 stabilizes RIPK2 in the cytosol by virtue reducing the polyubiquitination of the RIPK2 K48 residue, leading to homeostatic signalling upon NOD2 engagement. Therefore, an absence of YOD1 leads to the proteosomal degradation of RIPK2 resulting in a reduction of NOD2-mediated signalling upon NOD2 engagement, thereby contributing to increased colonic pathology following DSS treatment of mice.

Comments:

Prior to considering the limitations of the manuscript, I would like to commend the authors on several important aspects of their work:

- 1) The use of littermate controls within the in vivo YOD1-DSS experiments

- 2) The generation of the Yod1^{-/-} mice
- 3) The attempted mechanistic understanding of YOD1-mediated effects in intestinal inflammation using an array of in vitro and in vivo models

However, I recommend major revisions for this manuscript, and urge the authors to consider the following revisions for their manuscript:

Please indicate in each figure legend how many independent experiments the data are representative of.

- 1) Figure 1B - the staining does not look representative. The UC sample seems to have no staining for YOD1 when compared to the control, however this seems highly implausible given that the expression level of YOD1 in UC tissue is only marginally lower than the controls in Figure 1A. Quantification of multiple representative images is needed to make this stain convincing.
- 2) Figure 1M - The authors need to state the significance of the seemingly reduced number of goblet cells in the Yod1^{-/-} mice. I.e what does reduced amount of goblet cells mean in the context of intestinal inflammation? Why was this readout chosen instead of a conventional overall intestinal pathology scoring? Furthermore, a "representative image" is shown here. However, again, quantification of the goblet cell numbers between WT and YOD1 KO mice is needed using multiple images across each group once the authors have made it clear why they're examining the reduction in goblet cells.
- 3) Figure 2G - given the size of the images the F4/80 staining is impossible to see and judge. The image size needs to be increased and the intensity of the IHC stain needs to be increased.
- 4) Figure 2. The authors claim that YOD1 KO mice have increased macrophage infiltration in their colons following DSS. However, the only indication of this claim is the qPCR and faulty F4/80 data in figure 2. To make this claim, flow cytometry analysis of the resulting colonic leukocyte populations needs to be performed. This will allow for an exact quantitative difference in the amount of macrophages between WT and YOD1 KO mice. As it stands, qPCR data is nearly meaningless without protein-based analyses, and IHC staining is, at best, pseudo-quantitative.
- 5) Figure 3A and associated lines 328-329. You cannot make the claim that YOD1 is upregulated in colon-infiltrating macrophages. Given that inflammation induces more macrophages, the YOD1-staining will of course be more numerous. To make the claim that YOD1 is upregulated in colon-infiltrating macrophages you would need to sort the colonic macrophages and perform a qPCR.
- 6) Figure 3A: It is quite unclear what this IF staining is trying to depict, and the figure legend does not clarify. The figure legend states that this YOD1 WT and YOD1 KO F4/80 IF staining is shown, however it seems that only the "water" and "DSS" conditions are shown.
- 7) Figure 3. The figure legend states that "...deletion of YOD1 in macrophages exacerbates colitis". This is wrong. Your bone marrow chimera experiments show that a pan-hematopoietic deletion of YOD1 exacerbates colitis. To make this claim, you require the use of macrophage-specific cre and Yod1-flox experiments.
- 8) Figure 3N. Again, please quantify the difference in goblet cells.
- 9) Figure 4. The results in this figure seem completely incongruous with the in vivo qPCR data in Figures 2A-F. Why does deletion of YOD1 in BMDMs result in decreased Il-1b, IL-6, and TNF expression whereas in in vivo DSS-colitis models YOD1 KO mice produce more of these cytokines? An explanation is needed.
- 10) Figure 5H - please indicate how many biological replicates comprise these data. Is this an n=1? If so, further replicates are required.
- 11) Figure 7G-H. Again, please quantify and indicate why PAS/AB staining is a desired readout.
- 12) Lines 345-348. You cannot "rule out" the effect of IEC-intrinsic YOD1 signalling in ameliorating colitis by using an in vitro assay. Either rephrase to "decreases the possibility..." or generate IEC-specific Cre x Yod1 flox mice.

Referee #1:

The manuscript by Shen et al. describes the activity of Yod1, a K48 and K63 deubiquitinase on the pathogenesis of a mouse model of Ulcerative colitis-like pathology. Yod1 has previously been shown to be involved in multiple innate immune signaling pathways, including IL-1 (Shimmack et al., eLife, 2017) and MAVS (Lui et al, J. Immunology, 2019) and importantly, shows a well-published role in ER dynamics (Ernst et al., Mol Cell, 2009; Bernardi et al., Mol.Bio.Cell, 2013; etc). In the current work, Shen et al. link the DSS-colitis effect to the K48 deubiquitination of the scaffolding protein Kinase RIPK2, a member of the innate immune NOD1/2 signaling system. They then claim that Yod1 regulates RIPK2 stability.

These findings would be of interest to the innate immune community, but as submitted, the authors really fail to link Yod1 and RIPK2 together. They don't rule out ER involvement nor do they rule out IL-1 or MAVS involvement. In fact, the data linking Yod1 to RIPK2 are unconvincing and if anything, to this reviewer, don't show an effect of Yod1 to RIPK2 stability. For these reasons, the work seems preliminary and somewhat overstated and for this reason, enthusiasm is muted. My specific comments are as follows:

1. The connection here between Yop1 and RIPK2 is tenuous. The signal defect with L18-MDP in Figure 5A is minimal at best. Figure 5c shows loss of phospho-274 RIPK2, but this has not been consistently seen as MDP-inducible. Most labs use RIPK2

ubiquitination and/or XIAP binding as more convincing evidence of RIPK2 activity. In fact in Figure 5C, there's no difference in the half-life of RIPK2 upon L18-MDP stimulation in Yop1-null cells and then amounts of RIPK2 present are nearly identical in the Yod1-WT and Yod1-null BMDMs and in Figure 5D, the amounts of RIPK2 present don't seem very different - as quantified in Figure 5D, its unlikely this small of a difference causes a biological effect. The half-life differences in Figure 5G needs to be performed +/- MG132 (see transfection level concerns below) and the graph in 5H isn't really convincingly different. In Figure 6D, the amount of K48 ubiquitination is not noticeably different in a biological sense. The only experiment really linking the NOD2/RIPK2 pathway with Yod1 is the MDP-DSS protection experiment in Figure 7, but MDP or any other innate immune stimulus - esp LPS - is known to protect against DSS colitis. RIPK2, itself, has given variable results in DSS colitis assays with most studies showing little or no effect of RIPK2-WT or RIPK2-null mice in DSS assays.

Response: Figure 5A shows that deletion of YOD1 inhibited NOD2 signaling. Clear difference can be seen from the results, which is consistent with a previous report showing that degradation of RIPK2 by the E3 ligase ZNRF4 inhibits NOD2 signaling (Bist et al., *Nat Commun.* 2017).

We agree with the reviewer that phosphorylation of RIPK2 is dispensable for NOD2 signaling. As suggested by the reviewer, we performed new experiments. Deletion of YOD1 reduced RIPK2 protein levels but did not affect the binding between RIPK2 and XIAP (new Figure 5C). In addition, we performed another experiment to show that the K63 ubiquitination of RIPK2, which is essential for NOD2 signaling, was not changed by YOD1 (new Figure 6E).

As requested by the reviewer, we treated cells with or without MG132, and found that MG132 treatment increased RIPK2 protein abundance in *Yod1*^{+/+} and *Yod1*^{-/-} primary macrophages and blunted the difference between the two genotypes (new Figure 6F, input). We are sorry that the half-life experiment could not be performed in the presence of MG132, which inhibits protein degradation.

For figure 5H, we quantified results from more experiments and added the new quantification to the new Figure 5H.

As suggested by the reviewer, we performed new DSS colitis experiments with LPS pretreatment. Unlike MDP stimulation, which ameliorated colitis in *Yod1*^{+/+} mice, LPS stimulation ameliorated colitis in both *Yod1*^{+/+} and *Yod1*^{-/-} mice but could not remove the difference in disease severity

between the two mouse strains (new Figure EV1), showing a specific function of YOD1 in NOD2 signaling. Moreover, we found that the *in vivo* knockdown of RIPK2 exacerbated colitis in *Yod1*^{+/+} mice, and *Yod1*^{+/+} and *Yod1*^{-/-} mice showed similar disease severity after RIPK2 knockdown (new Figure 8).

The reviewers raised concerns about the differences in our experiments. As fine-tuning mechanism, DUBs could not completely remove ubiquitination from the substrates. We could not get results with black-and-white differences in our experiments. However, the difference in RIPK2 protein abundance is big enough to affect biological activities (Bist et al., *Nat Commun.* 2017).

On top of the above, Yop1 has been shown to interact with and affect TRAF6 - a major player in TLR and IL-1 signaling and has been shown to affect ER dynamics. In my mind, it's far more likely that Yop1 has a more general role in innate immune signaling - most likely at a cell biological level - than a specific effect on RIPK2, especially given the data shown in Figures 3 and 6. Lastly, RIPK2 ubiquitination is known to be regulated by other deubiquitinases including A20 and CYLD (Abbott et al., *Current biology*, 2004; Hito Sumatsu et al., *Immunity* 2008). It's likely that there's redundancy in RIPK2 deubiquitination and for this reason, the singular effect of Yod1 needs far more proof than what is provided.

Response: Indeed, YOD1 has been shown to inhibit IL-1 signaling in Schimmack et al., *Elife*. 2017, which has been cited in our manuscript. However, our own data show that deletion of YOD1 had no effect on LPS-induced pro-inflammatory responses in primary macrophages (new Figure S9A-C). Besides, in another unpublished study, we found that *Yod1*^{+/+} and *Yod1*^{-/-} primary microglia produced comparable levels of cytokines upon LPS stimulation. In the current study, we also found that LPS pretreatment protected DSS-induced colitis in both *Yod1*^{+/+} and *Yod1*^{-/-} mice but could not blunt the differences between the two genotypes (new Figure EV1), further consolidating that YOD1 has no effect on LPS-induced signaling.

We agree with the reviewer that YOD1 is not the only DUB that can deubiquitinate RIPK2, which has been shown to be deubiquitinated by CYLD, MYSM1, and A20. However, these DUBs regulate non-degradative ubiquitination of RIPK2 and we show that YOD1 can stabilize RIPK2 through deubiquitination. These DUBs and E3s have been cited and discussed in the Discussion (Lines 286-295).

2. Continuing on point 1 above, but from a technical standpoint, there are a number of

issues with the molecular biology in the manuscript. In Figure 4, RAW macrophages are notoriously difficult to transfect and can initiate an innate immune response to intracellular DNA when transfected. Good transfection efficiencies are around 25% (see Cheung et al JOVE, 2015 for a good summary) and there's considerable cellular response to DNA. Given that, it is not surprising that Figure 5G shows minimal results when FLAG-Yod1 is transfected. In these types of experiments, one would typically make an inducible Yod1 in a stably transduced viral system to overcome transfection efficiencies and effects. Additionally, in Figure 6, when looking at ubiquitinated proteins, one typically needs to denature the lysate before performing the IP. This is to rid the target protein of ubiquitinated binding partners - these can give artifactual findings. This is particularly important with RIPK2 as it forms a complex with proteins known to be ubiquitinated including XIAP and NEMO. Figure 6D needs to do this. Lastly, the ubiquitination experimentation is underdone. Yod1 appears to have K63 and K48 deubiquitinase activity. Does Yod1 deubiquitinate K63-linked RIPK2? What is K48-ubiquitinating RIPK2 and does Yod1 interact with that? Half life experiments would be better performed with MDP-stimulation - are the authors supposing that Yod1 regulates basal, unstimulated RIPK2? RIPK2 expression is known to be induced by inflammatory stimuli like LPS or TNF (Kobayashi et al., Nature, 2005). Does this induction affect Yod1 activity on RIPK2? Is Yod1 specific to RIPK2? Does it affect TRAF6 as well and could this underlie some of the results seen?

Response: We agree with the reviewer that RAW264.7 cells were not optimal for the experiment. Therefore, we performed new experiments with primary macrophages. As shown in the new Figure 5G, deletion of YOD1 significantly accelerated RIPK2 degradation in the CHX chase experiment.

As suggested by the reviewer, we performed new IP experiments under denaturing conditions according to previously reported methods (Panda., *Immunity*. 2015; Panda et al., *Nat Commun.* 2018). The new results are included in the new Figure 6E-F.

K63 ubiquitination of RIPK2 is critical for NOD2 signaling and it is catalyzed by E3s including TRAF6, XIAP, ITCH, RNF186 and Pellino3, and inhibited by DUBs including CYLD, MYSM1, and A20. K63 ubiquitination can also mark proteins for degradation in the lysosome. However, in our new IP experiment after denaturation (new Figure 6E), YOD1 did not affect K63 ubiquitination of RIPK2. These results are consistent with the findings that YOD1 inhibited the proteasomal degradation of RIPK2 via K48 deubiquitination (new Figure 6D-G).

As requested by the reviewer, we additionally performed CHX chase experiment with MDP. Deficiency of YOD1 significantly accelerated the degradation of RIPK2 under MDP stimulation (new Figure 5G).

To test whether the activity of YOD1 on RIPK2 is regulated by LPS or TNF, we performed new IP experiments. Stimulation of macrophages with LPS or TNF did not increase the binding between YOD1 and RIPK2 (new Figure S13A). In contrast, L18-MDP stimulation increased the interaction between YOD1 and RIPK2 (new Figure S13B). In addition, YOD1 deletion had no impact on pro-inflammatory gene production and signal transduction in response to TNF or LPS, further demonstrating that YOD1 is dispensable for LPS- and TNF-induced inflammatory responses (new Figure S9A-F). Although YOD1 has been shown to regulate IL-1 signaling by interacting with TRAF6, we could not observe this effect in the present study. Here, we have to honestly report this observation.

3. RAW264.7 cells are not the best in which to study a NOD2:RIPK2 signaling axis.

People typically use PMA-differentiated THP-1 cells or immortalized BMDMs for biochemical experiments as the signaling is much stronger. This may be one reason the results in Figure 5 aren't robust.

Response: Actually, primary macrophages were used for analyzing signal transduction in Figure 5. In the present study, RAW264.7 cells were only used to validate results obtained with primary macrophages. In addition, to minimize the effect of transfected DNA on NOD signaling, we transfected vector plasmids in the control group.

4. Given all this, I find it far more likely that Yod1 affect innate immunity broadly. The authors should test LPS, PamCys, PMA, IL-1, etc on the BMDMs and see if there's a more broad innate immune effect. Figure 4, for instance, would benefit from comparison with other innate immune stimuli. "Relative expression" really doesn't give the reader a sense

of levels or effect. These levels should be formally tested by ELISA. MDP likely gives a 100-fold lower effect than LPS.

Response: As suggested by the reviewer, we stimulated *Yod1*^{+/+} and *Yod1*^{-/-} primary macrophages with PMA, LPS, L18-MDP, and TNF- α . ELISA analysis of the supernatant showed that YOD1 deficiency specifically reduced IL-6 production in response to L18-MDP (new Figure S9C). In addition, signal transduction and pro-inflammatory gene transcription were not altered by YOD1 deletion in response to TNF, a detrimental cytokine in human IBD (new Figure S9D-F). Together, these results show that YOD1 specifically enhances NOD2 signaling in primary macrophages.

5. Within the field, there hasn't really been a phenotype of RIPK2-null mice with DSS.

Most experiments have needed to use TNBS, Samp/YitFc, TRUC mice on Balb/C background to see an effect. There's also the well-known role of the microbiota on DSS experiments and on NOD2 pathology in general. These don't seem to have been accounted for in the experimentation. At the very least, the authors should use co-housed mice with RIPK2 inhibitors and see if there's an additive effect on the *Yod1*-null mice. Additionally mating RIPK2-null mice with *Yod1* mice could be performed to see if there's an additive effect.

Response: We agree with the reviewer that the microbiota significantly affect DSS colitis. Therefore, all the mice were co-housed for one week before DSS treatment to ensure that mice have similar microbiota. We have specified this experimental detail in the Methods (Lines 362-363).

Most of RIPK2 inhibitors inhibit the kinase activity of RIPK2, which is not required for NOD2 signaling. In addition, inhibitors may have off-target effects. Therefore, we did not use RIPK2 inhibitors. Instead, we used AAV vectors to silence RIPK2 in *Yod1*^{+/+} and *Yod1*^{-/-} mice. Knockdown of RIPK2 exacerbated colitis in *Yod1*^{+/+} mice, and *Yod1*^{+/+} and *Yod1*^{-/-} mice had similar disease severity after RIPK2 knockdown (new Figure 8).

6. It's well-known that innate immune priming (typically with LPS, but also with MDP) can protect against DSS colitis. A LPS-protective effect on DSS colitis should still be seen in the *Yod1* mice.

Response: As suggested by the reviewer, we have performed additional experiments with LPS pretreatment. As shown in the new Figure EV1, low-dose LPS treatment ameliorated DSS colitis in both *Yod1*^{+/+} and *Yod1*^{-/-} mice but could not blunt the differences between the two genotypes, indicating that YOD1 regulates other signaling pathways.

7. Figure 7 needs to look at RIPK2 ubiquitination and RIPK2 levels in the mice. RIPK2 levels should be much lower in the *Yod1*-null mice.

Response: As suggested by the reviewer, we analyzed the ubiquitination status and protein abundance of RIPK2 in the colon of *Yod1*^{+/+} and *Yod1*^{-/-} mice. As shown in the new Figure S10A-B, the reduced RIPK2 protein levels in colons of *Yod1*^{-/-} mice are accompanied with increased K48 ubiquitination.

In summary, I find the link between *Yod1* and RIPK2 to be tenuous. *Yod1* is more likely affecting the innate immune system more broadly - like through ER dynamics, but also through IL-1 and MAVS signaling (as published). RIPK2 may be playing a role here, but other innate immune pathways may predominate, especially as RIPK2's role in DSS colitis has been variable across labs. Given these concerns, coupled with the molecular biology issues outlined above, enthusiasm is muted.

Response: We thank the reviewer for the careful evaluation of our manuscript. It is undeniable that TLR4 and other innate immune signaling pathways play a critical role in colitis. However, loss-of-function mutations in NOD2 are conclusively linked with human IBD. In this study, we experimentally confirmed the regulatory function of YOD1 in colitis by stabilizing RIPK2. As mentioned by the reviewer "RIPK2's role in DSS colitis has been variable across labs", we found that YOD1 did not regulate LPS-induced inflammatory responses in primary macrophages and microglia (unpublished data) derived from *Yod1*^{+/+} and *Yod1*^{-/-} mice, although YOD1 has been shown to inhibit IL-1 signaling. Nevertheless, we cited this paper to inspire further studies with a more solid conclusion. In addition, we found that silencing of RIPK2 significantly exacerbated DSS colitis in mice (new Figure 8), which is consistent with the established role of NOD2 signaling in intestinal inflammation.

References

[1] Bist P, Cheong WS, Ng A, et al. E3 Ubiquitin ligase ZNRF4 negatively regulates NOD2 signalling and induces tolerance to MDP. *Nat Commun.* 2017;

[2] Schimmack G, Schorpp K, Kutzner K, et al. YOD1/TRAF6 association balances p62-dependent IL-1 signaling to NF- κ B. *Elife*. 2017;

[3] Panda S, Nilsson JA, Gekara NO. Deubiquitinase MYSM1 Regulates Innate Immunity through Inactivation of TRAF3 and TRAF6 Complexes. *Immunity*. 2015;

[4] Panda S, Gekara NO. The deubiquitinase MYSM1 dampens NOD2-mediated inflammation and tissue damage by inactivating the RIP2 complex. *Nat Commun*. 2018;

Referee #2:

Shen et al. discover that patients with ulcerative colitis exhibit elevated levels of YOD1. Their study suggests that YOD1 plays a regulatory role in the NOD2 pathway. Through in vivo experiments involving bone marrow transplants, the authors convincingly demonstrate that immune cells lacking YOD1 exacerbate inflammation following DSS colitis. This finding underscores the protective role of YOD1. Additionally, the study highlights a reduction in YOD1 expression alongside an increase in F4/80 cell infiltration in both DSS-treated mice and ulcerative colitis patients. Consistent with existing literature, activating the NOD2 pathway via MDP treatment conveys protective effects in DSS-treated mice, this protection is significantly diminished by YOD1 deficiency. The authors employ a range of cell biology and biochemistry techniques to link YOD1 deficiency with decreased NOD2-dependent NF- κ B signalling and cytokine production. They attribute these outcomes to lowered RIPK2 levels, positing that this is due to heightened K48-ubiquitination of RIPK2 in the absence of YOD1's catalytic activity.

The in vivo data presented is of high quality, and the elucidated role of YOD1 in

DSS-colitis will be of interest to those in the NOD2 research community, hinting at a novel therapeutic target for managing NOD2-driven ulcerative colitis. However, the paper should clarify whether YOD1's effect on NF- κ B signalling occurs via a TRAF6/TNF-dependent or independent pathway. While the mechanistic insights (from cell biology and biochemistry studies) mostly suggest correlation, further evidence is needed to confirm that YOD1 directly influences the NOD2 pathway, and that this influence is the driving force for the in vivo phenotype.

Crucial Concerns:

- Referencing Stafford et al. Cell Reports (2018), which demonstrates the interplay between NOD2 and the TNF pathway in cytokine response, the paper should address whether TRAF6 and TNFR1 might account for the observed effects, given YOD1's known regulatory impact on TRAF6. Additionally, given RIPK2's precise protein level regulation, any basal changes in NF- κ B signalling in YOD1 knockouts could influence RIPK2 levels, and therefore NOD2 signalling. Looking at Fig 5C, it appears that NOD2 levels might be increased in the KO, hinting towards such an event as NOD2 is also sensitive to NF- κ B changes.
- Although interactions between YOD1 and RIPK2 are noted, this does not rule out YOD1's potential impact on other pathways. The observed phenotypes and weak biochemical evidence regarding YOD1's influence on RIPK2's K48 ubiquitination call for more robust investigation.
- The authors reference Bist et al, 2017 and the K48 ubiquitination of RIPK2. In that study,

they indicate RIPK2 is K48-ubiquitinated following MDP treatment, rather than basal RIPK2 regulation. In this study, there is a lack of investigation into whether YOD1 impacts basal or NOD2 activated RIPK2 levels, especially considering that a major finding of the study is MDP treatment protects DSS-colitis in an MDP dependent manner.

Suggested Experiments:

1. To verify that the NOD2 cytokine phenotype stems from YOD1's action on RIPK2 rather than the TNF pathway, compare cytokine responses in WT and YOD1 KO BMDMs and primary cells from the peritoneum of mice, with and without a TNF blocking antibody, upon NOD2 activation.

Response: We thank the reviewer for this constructive suggestion. Indeed, TNF is a detrimental cytokine in colitis and anti-TNF therapy is effective in clinical treatment for IBD. To confirm whether YOD1 affects cytokine production by regulating TNF signaling, we stimulated *Yod1*^{+/+} and *Yod1*^{-/-} primary macrophages with TNF. As shown in the new Figure S9C-F, YOD1 deletion did not change TNF-induced signal transduction and cytokine production, providing a direct evidence that YOD1 does not regulate TNF signaling. In addition, we stimulated *Yod1*^{+/+} and *Yod1*^{-/-} primary macrophages with PMA, LPS, L18-MDP, and TNF. ELISA analysis of the supernatant found that YOD1 deficiency specifically reduced IL-6 production in response to L18-MDP (new Figure S9C). Although YOD1 has been shown to inhibit IL-1 signaling by regulating TRAF6, we found that YOD1 did not regulate LPS-induced inflammatory responses in primary macrophages and microglia (unpublished data) derived from *Yod1*^{+/+} and *Yod1*^{-/-} mice.

2. Investigate whether the TNF pathway significantly contributes to the observed differences in WT versus YOD1 BMDM responses to TNF treatment (TNF timecourse and Western blotting of NF-κB/MAPK activation).

Response: As suggested by the reviewer, we stimulated *Yod1*^{+/+} and *Yod1*^{-/-} primary macrophages with TNF for different time points and then performed Western blot analysis of signaling molecules. In line with cytokine production results (new Figure S9C-E), YOD1 deletion did not affect the activation of NF-κB/MAPK upon TNF treatment (new Figure S9F), ruling out the regulatory function of YOD1 in TNF signaling.

3. Strengthen the evidence that YOD1 directly affects K48 ubiquitination on RIPK2 through specific experiments designed to isolate this interaction in a time dependent manner (+/- MG132, +/- MDP treatment, Tandem Ubiquitin Binding Entities TUBEs, then blot for RIPK2 to ensure ubiquitination is directly linked to RIPK2).

Response: As requested by the reviewer, we treated *Yod1*^{+/+} and *Yod1*^{-/-} primary macrophages +/- L18-MDP or MG132. Thereafter, we denatured the lysates before performing IP according to previously reported methods (Panda., *Immunity*. 2015; Panda et al., *Nat Commun*. 2018) to detect ubiquitination directly linked to RIPK2. As shown in the new Figure 6E-F, YOD1 was not detected in the IP samples, showing that the denaturation was successful. The new results show that L18-MDP or MG132 treatment increased K48 ubiquitination of RIPK, and YOD1 deletion strongly increased RIPK2 K48 ubiquitination (new Figure 6E-F).

Minor concerns:

- Improve and strengthen evidence and clarity in Figure 2g's depiction of F4/80 staining and macrophage infiltration in WT vs. YOD1 KO mice. Ensure arrows clearly point to the relevant cells and provide higher-quality images.

Response: As suggested by the reviewer, we performed new experiments to provide images of high quality (new Figure 2G). In addition, we isolated infiltrating leukocytes from colons and performed flow cytometry to analyze colonic macrophages. As shown in the new Figure 2H-I, more macrophages were detected in colons of *Yod1*^{-/-} mice.

- Clarify whether mass spectrometry results from pulldowns (hit list), will be published, and whether TRAF6 was shown to interact.

Response: The mass spectrometry result will be published together with this manuscript. However, TRAF6 was not detected in the interactome of YOD1.

- For further validation, include all ladder markers in the raw and published Western blots in both the review process and final manuscript.

Response: As requested by the reviewer, we have added ladder markers for all Western blot images.

- Correct the discussion on the role of cIAP1/2 and XIAP in RIPK2 ubiquitination, referencing Stafford et al. (2018) to accurately represent XIAP's primary role over cIAP1/2 for K63 linked ubiquitination of RIPK2.

Response: We agree with the reviewer that XIAP is more important than cIAP1/2 in K63 ubiquitination of RIPK2. We have replaced cIAP1/2 with XIAP and cited the paper Stafford et al. (2018) as a reference (Lines 286-291).

Addressing these points will significantly enhance the manuscript's contribution to the field, offering a clearer and more compelling narrative of YOD1's role in regulating the NOD2 pathway in ulcerative colitis.

References

- [1] Panda S, Nilsson JA, Gekara NO. Deubiquitinase MYSM1 Regulates Innate Immunity through Inactivation of TRAF3 and TRAF6 Complexes. *Immunity*. 2015;
- [2] Panda S, Gekara NO. The deubiquitinase MYSM1 dampens NOD2-mediated inflammation and tissue damage by inactivating the RIP2 complex. *Nat Commun*. 2018;

Referee #3:

The manuscript "YOD1 sustains NOD2-mediated protective effects in colitis by stabilizing RIPK2" by Shen and colleagues explores a hitherto unknown regulator of intestinal inflammation. The authors first describe an aberrant colonic expression pattern of the deubiquitinase, YOD1, in ulcerative colitis (UC) and during murine DSS-induced colonic

inflammation. Then, the group generated Yod1-deficient mice and subjected them to an acute stimulus of DSS and compared their colonic pathologies relative to littermate wildtype (WT) mice. The group found that YOD1-deficiency resulted in a greater intestinal inflammatory burden in various readouts of intestinal inflammation. Using a sequence of gene transcription, bone marrow chimera, and immunostaining experiments the authors suggest that YOD1-deficiency specifically in macrophages drives this increased inflammation. Specifically, the authors suggest that YOD1-deficiency induces increased macrophage infiltration into the colons of DSS-treated mice, which is associated with an increase in colonic proinflammatory gene transcription. To explore the macrophage-intrinsic role of YOD1, the authors treated WT and YOD1 KO BMDMs with the synthetic NOD2-agonist, L18-MDP, and found that YOD1-deficiency lead to decreased proinflammatory gene transcription following NOD2 agonism. Mass-spectrometry and Co-IP experiments suggest that YOD1 physically interacts with RIPK2; a downstream regulator of NOD2-mediated signalling. More specifically, the authors suggest that YOD1 stabilizes RIPK2 in the cytosol by virtue reducing the polyubiquitination of the RIPK2 K48 residue, leading to homeostatic signalling upon NOD2 engagement. Therefore, an absence of YOD1 leads to the proteosomal degradation of RIPK2 resulting in a reduction of NOD2-mediated signalling upon NOD2 engagement, thereby contributing to increased colonic pathology following DSS treatment of mice.

Comments:

Prior to considering the limitations of the manuscript, I would like to commend the authors on several important aspects of their work:

- 1) The use of littermate controls within the in vivo YOD1-DSS experiments
- 2) The generation of the Yod1^{-/-} mice
- 3) The attempted mechanistic understanding of YOD1-mediated effects in intestinal inflammation using an array of in vitro and in vivo models

However, I recommend major revisions for this manuscript, and urge the authors to consider the following revisions for their manuscript:

Please indicate in each figure legend how many independent experiments the data are representative of.

Response: As requested by the reviewer, we have included the number of independent experiments in the figure legend.

1) Figure 1B - the staining does not look representative. The UC sample seems to have no staining for YOD1 when compared to the control, however this seems highly implausible given that the expression level of YOD1 in UC tissue is only marginally lower than the controls in Figure 1A. Quantification of multiple representative images is needed to make this stain convincing.

Response: As suggested by the reviewer, we have replaced old images with more representative images and added quantification of multiple images (new Figure 1B).

2) Figure 1M - The authors need to state the significance of the seemingly reduced number of goblet cells in the Yod1^{-/-} mice. I.e what does reduced amount of goblet cells mean in the context of intestinal inflammation? Why was this readout chosen instead of a conventional overall intestinal pathology scoring? Furthermore, a "representative image" is shown here. However, again, quantification of the goblet cell numbers between WT and YOD1 KO mice is needed using multiple images across each group once the authors have made it clear why they're examining the reduction in goblet cells.

Response: Goblet cells are specialized epithelial cells that secrete mucus and anti-microbial proteins, playing a critical role in the barrier function and innate immunity of the intestine. Under conditions of intestinal inflammation, increased loss of goblet cells is associated with enhanced pathology. Therefore, the number of goblet cells is widely used to evaluate intestinal damage. As requested by the reviewer, we have added the quantification of goblet cells in the new Figure S4.

3) Figure 2G - given the size of the images the F4/80 staining is impossible to see and judge. The image size needs to be increased and the intensity of the IHC stain needs to be increased.

Response: As suggested by the reviewer, we have replaced the images with new images of better quality (new Figure 2G).

4) Figure 2. The authors claim that YOD1 KO mice have increased macrophage infiltration in their colons following DSS. However, the only indication of this claim is the qPCR and faulty F4/80 data in figure 2. To make this claim, flow cytometry analysis of the resulting colonic leukocyte populations needs to be performed. This will allow for an exact quantitative difference in the amount of macrophages between WT and YOD1 KO mice. As it stands, qPCR data is nearly meaningless without protein-based analyses, and IHC

staining is, at best, pseudo-quantitative.

Response: As suggested by the reviewer, we isolated infiltrating leukocytes from colons of *Yod1*^{+/+} and *Yod1*^{-/-} mice, and then performed flow cytometry to analyze colonic macrophages. As shown in the new Figure 2H-I, more macrophages were detected in colons of *Yod1*^{-/-} mice.

5) Figure 3A and associated lines 328-329. You cannot make the claim that YOD1 is upregulated in colon-infiltrating macrophages. Given that inflammation induces more macrophages, the YOD1-staining will of course be more numerous. To make the claim that YOD1 is upregulated in colon-infiltrating macrophages you would need to sort the colonic macrophages and perform a qPCR.

Response: We isolated macrophages from the colon by F4/80 magnetic beads and detected *Yod1* transcription in macrophages by qRT-PCR. As shown in the new Figure S5, *Yod1* mRNA levels were upregulated in colonic macrophages of mice suffering from colitis.

6) Figure 3A: It is quite unclear what this IF staining is trying to depict, and the figure legend does not clarify. The figure legend states that this YOD1 WT and YOD1 KO F4/80 IF staining is shown, however it seems that only the "water" and "DSS" conditions are shown.

Response: We are sorry for this mistake. We have corrected the figure legend as "Representative YOD1 (red) and F4/80 (green) immunofluorescence staining of the colon from control and DSS-treated mice on day 8", lines 720-721.

7) Figure 3. The figure legend states that " ...deletion of YOD1 in macrophages exacerbates colitis". This is wrong. Your bone marrow chimera experiments show that a pan-hematopoietic deletion of YOD1 exacerbates colitis. To make this claim, you require the use of macrophage-specific cre and *Yod1*-flox experiments.

Response: We agree with the reviewer and changed "Deletion of YOD1 in macrophages" to

“pan-hematopoietic deletion of YOD1”, line 719. Accordingly, we also changed the statement in the text, lines 128, 155.

8) Figure 3N. Again, please quantify the difference in goblet cells.

Response: We have added quantification for goblet cells (new Figure S7).

9) Figure 4. The results in this figure seem completely incongruous with the in vivo qPCR data in Figures 2A-F. Why does deletion of YOD1 in BMDMs result in decreased Il-1b, IL-6, and TNF expression whereas in in vivo DSS-colitis models YOD1 KO mice produce more of these cytokines? An explanation is needed.

Response: NOD2-induced physiological inflammation keeps intestinal microbes under control, which is critical for the maintenance of intestinal homeostasis. Impairment in the NOD2 signaling renders the host unable to control intestinal microbes, resulting in intestinal inflammation. In humans, loss-of-function mutations in NOD2 are conclusively linked with IBD (Ogura et al., *Nature*. 2001; Hugot et al., *Nature*. 2001; Hampe et al. *Lancet*. 2001). Therefore, in the intestine, the NOD2 signaling, which initiates protective physiological inflammation, is totally different from TLR4 signaling, which triggers detrimental pathological inflammation. Our results are consistent with previous reports (Bertrand et al., *Immunity*. 2009; Yang et al., *Nat Immunol*. 2013)

10) Figure 5H - please indicate how many biological replicates comprise these data. Is this an n=1? If so, further replicates are required.

Response: In the figure, representative images are shown. We have added “n” and new statistics of multiple biological replicates for this result (new Figure 5H).

11) Figure 7G-H. Again, please quantify and indicate why PAS/AB staining is a desired readout.

Response: We have added quantification for goblet cells (new Figure S15).

12) Lines 345-348. You cannot "rule out" the effect of IEC-intrinsic YOD1 signalling in

ameliorating colitis by using an in vitro assay. Either rephrase to "decreases the possibility..." or generate IEC-specific Cre x Yod1 flox mice.

Response: We thank the reviewer for pointing out this problem. As suggested, we have rephrased it to "decreases the possibility", line 153.

References

- [1] Ogura Y, Bonen DK, Inohara N, et al. A frameshift mutation in NOD2 associated with susceptibility to Crohn's disease. *Nature*. 2001;
- [2] Hugot JP, Chamaillard M, Zouali H, et al. Association of NOD2 leucine-rich repeat variants with susceptibility to Crohn's disease. *Nature*. 2001;
- [3] Hampe J, Grebe J, Nikolaus S, et al. Association of NOD2 (CARD 15) genotype with clinical course of Crohn's disease: a cohort study. *Lancet*. 2002;
- [4] Bertrand MJ, Doiron K, Labbé K, Korneluk RG, Barker PA, Saleh M. Cellular inhibitors of apoptosis cIAP1 and cIAP2 are required for innate immunity signaling by the pattern recognition receptors NOD1 and NOD2. *Immunity*. 2009;
- [5] Yang S, Wang B, Humphries F, et al. Pellino3 ubiquitinates RIP2 and mediates Nod2-induced signaling and protective effects in colitis. *Nat Immunol*. 2013;

Dear Prof. Wang,

Thank you for the submission of your revised manuscript to our editorial offices. I have now received the reports from the three referees that I asked to re-evaluate the study, you will find below. As you will see, the referees now fully support the publication of the study in EMBO reports. Referees #1 and #3 have suggestions or requests to improve the manuscript, I ask you to address in a final revised manuscript. Please also provide a final p-b-p-response addressing these points.

- I would suggest a more comprehensive title:

YOD1 sustains NOD2-mediated physiological inflammation in colitis by stabilizing RIPK2

or

YOD1 sustains NOD2-mediated protective signalling in colitis by stabilizing RIPK2

- Please have your final manuscript carefully proofread by a native speaker. The text still contains typos and grammatical errors.

- Please provide the abstract written in present tense throughout.

- Please order the manuscript sections like this, using these names:

Title page - Abstract - Keywords - Introduction - Results - Discussion - Methods - Data availability section - Acknowledgements - Disclosure and Competing Interests Statement - References - Figure legends - Expanded View Figure legends

- Please make sure that the number "n" for how many independent experiments were performed, their nature (biological versus technical replicates), the bars and error bars (e.g. SEM, SD) and the test used to calculate p-values is indicated in the respective figure legends. Please also check that all the p-values are explained in the legend, and that these fit to those shown in the figure. Please provide statistical testing where applicable. Please avoid the phrase 'independent experiment', but clearly state if these were biological or technical replicates. Please also indicate (e.g. with n.s.) if testing was performed, but the differences are not significant. In case n=2, please show the data as separate datapoints without error bars and statistics. See also: <http://www.embopress.org/page/journal/14693178/authorguide#statisticalanalysis>

If n<5, please show single datapoints for diagrams. It seems for several diagrams n.s. is not indicated (in the main and Appendix figures). Moreover:

- Please note that the figure and figure legends for panels I-L of Fig. 3 are missing in the manuscript (but there are panels M and N). We are not sure if these figures as well as their legends are missing or if there is an issue with labelling. Please check.

- Please note that the exact p values are not provided in the legends of figures 1a-d, f, h-l; 2a-f, i; 3d-f, h, m; 4a-f; 5e, g-h; 7a-c, e-f; 8c-e, g-i; EV 1b-d, f-h.

- Please define the error bars in the legend of figure 1b.

- Please remove the sentence 'Source data are available online for this figure' from the legends. SD will be directly linked to the figures in the online version of the paper, thus this will be obvious.

- Please add scale bars of similar style and thickness to all microscopic images (main, EV and Appendix figures), using clearly visible black or white bars (depending on the background). Please place these in the lower right corner of the images themselves. Please do not write on or near the bars in the image but define the size in the respective figure legend. Presently, most scale bars are too thin. Please improve.

- Please make sure that all the funding information is also entered into the online submission system and that it is complete and similar to the one in the acknowledgement section of the manuscript text file. Presently, the grant from the Natural Science Foundation of Zhejiang Province (LZ24H090003) is missing in the submission system. Please check.

- Please make sure that all figure panels are called out separately and sequentially. It seems, presently a panel 3D and individual callouts for the panels of Fig. 4 are missing. Please check.

- Please make sure that the Appendix figures are named Appendix Fig. Sx and called out using this name. There is one callout for "Supplementary Figure 2" in the manuscript text. Please check.

- There are three Excel files uploaded ZIPed together (RAW1_LPS IgG.xlsx, RAW2_Con YOD1 IP.xlsx, and RAW3_LPS YOD1.xlsx). I assume this refers to Appendix Table S1 ("Appendix Table S1 - The data are available online as EXCEL files"). Please upload these files as separate datasets using the filename Dataset EVx. Please put a legend for these on the first TAB. Moreover, please add callouts for these files to the manuscript text.

- Please remove the present text regarding Appendix Table S1 from the Appendix ("Appendix Table S1 - The data are available

online as EXCEL files."). As indicated above, these Excel files need to be uplidd as Datasets EV1, EV2 and EV3. Then rename the Appendix table accordingly, i.e. Appendix Table S2 will become S1 and Appendix Table S3 will become S2. Finally, please update their callouts.

- Thank you for providing the requested source data. Please upload this as one folder per main figure (with all files for one figure in one folder and ZIPed) and one folder for EV and Appendix figure.

- Please note that all corresponding authors are required to supply an ORCID ID for their name upon submission of a revised manuscript. Please do that for co-corresponding author Jing Ruan. Please find instructions on how to link the ORCID ID to the account in our manuscript tracking system in our Author guidelines:
<http://www.embopress.org/page/journal/14693178/authorguide#authorshipguidelines>

In addition, I would need from you uploaded separately:

Best,

Referee #1:

The authors have addressed most of my concerns. I do think there needs to be a paragraph in the discussion concerning how YOD1 has been shown to be involved in IL-1 and LPS (TRAF6) and why the authors think they are not seeing the same results. Otherwise, I think the concerns have been addressed.

Referee #2:

In this revised manuscript, Shen et al. have effectively addressed the questions raised during the review of their study, which identifies YOD1 as a novel regulator of colitis via the NOD2 signalling pathway. The manuscript has undergone substantial modifications, resulting in significant improvements. The key strengths of the revision include:

1. Enhanced histological analysis from in vivo studies, incorporating both mouse and patient data.
 2. Clearer delineation of macrophages as a driving factor in the observed phenotype.
 3. Inclusion of additional controls to rule out pathway cross-talk, ensuring that the YOD1 phenotype is attributed to NOD2 rather than other pathways like LPS/TNF.
 4. Impressive immunoprecipitation of the NOD2 complex, demonstrating that RIPK2 levels are affected by YOD1 activity, rather than the complex formation itself in response to NOD2 stimuli.
 5. Improved biochemical analysis of the increased K48 ubiquitination of RIPK2 in the absence of YOD1 under denatured conditions.
 6. Effective use of RIPK2 knockdown via AAV, showing that RIPK2 knockdown phenocopies the YOD1 knockout.
- Overall, these revisions strengthen the study, making it more robust. I am confident that the authors have successfully identified and validated an important regulator of the NOD2 signalling pathway, which may hold potential for therapeutic intervention.

Referee #3:

In their revised version, the authors of the manuscript "YOD1 sustains NOD2-mediated protective effects in colitis by stabilizing RIPK2" have addressed most of the initial concerns raised by this reviewer. There are however some minor corrections that should be addressed prior to publication.

Figure 2

2G: The F4/80 staining in the Yod1^{-/-} DSS condition does not look convincing given that the zoomed image amplifies an area of stain that is in the lumen of the colonic roll. This staining looks to be more like background than a real stain. Given that intestinal immune cells are mostly found in the lamina propria, the authors should zoom into an area where the viewer can see the lamina propria and a somewhat intact tissue architecture (as they have done in the Yod1^{-/-} condition).

2H: Please include a representative gating scheme for the flow cytometry contour plots.

2I: 1) The Y-axis is confusing. Is this a percentage of all cells in the colon? Viable cells? Singlets? Myeloid cells? Please clarify. Furthermore, including a representative gating scheme in the supplementary information as previously suggested also clarifies these concerns. 2) Spelling..."Neutrophils" not "Neutrophiles". Please correct.

Dear Prof. Wang,

Thank you for the submission of your revised manuscript to our editorial offices. I have now received the reports from the three referees that I asked to re-evaluate the study, you will find below. As you will see, the referees now fully support the publication of the study in EMBO reports. Referees #1 and #3 have suggestions or requests to improve the manuscript, I ask you to address in a final revised manuscript. Please also provide a final p-b-p-response addressing these points.

- I would suggest a more comprehensive title:

YOD1 sustains NOD2-mediated physiological inflammation in colitis by stabilizing RIPK2

or

YOD1 sustains NOD2-mediated protective signalling in colitis by stabilizing RIPK2

Response: Thank you for your suggestion. We have changed the title to “YOD1 sustains NOD2-mediated protective signaling in colitis by stabilizing RIPK2”.

- Please have your final manuscript carefully proofread by a native speaker. The text still contains typos and grammatical errors.

Response: We have carefully checked the manuscript and eliminated typos and grammatical errors.

- Please provide the abstract written in present tense throughout.

Response: The abstract is now in present tense.

- Please order the manuscript sections like this, using these names:

Title page - Abstract - Keywords - Introduction - Results - Discussion - Methods - Data availability section - Acknowledgements - Disclosure and Competing Interests Statement - References - Figure legends - Expanded View Figure legends

Response: We have ordered the manuscript in the requested order.

- Please make sure that the number "n" for how many independent experiments were performed, their nature (biological versus technical replicates), the bars and error bars (e.g. SEM, SD) and the test used to calculate p-values is indicated in the respective figure legends. Please also check that all the p-values are explained in the legend, and that these fit to those shown in the figure. Please provide statistical testing where applicable. Please avoid the phrase 'independent experiment', but clearly state if these were biological or technical replicates. Please also indicate (e.g. with n.s.) if testing was performed, but the differences are not significant. In case n=2, please show the data as separate datapoints without error bars and statistics. See also:

<http://www.embopress.org/page/journal/14693178/authorguide#statisticalanalysis>

If $n < 5$, please show single datapoints for diagrams. It seems for several diagrams n.s. is not indicated (in the main and Appendix figures). Moreover:

Response: We have included the number of experiments, the nature of “n”, and the statistical tests in the figure legends. Moreover, we have re-checked the p-values presented in figures and replaced asterisks with specific values. Single datapoints were included in diagrams.

- Please note that the figure and figure legends for panels I-L of Fig. 3 are missing in the manuscript (but there are panels M and N). We are not sure if these figures as well as their legends are missing or if there is an issue with labelling. Please check.

Response: Thank you for pointing out this mistake. We have changed “M-N” to “I-L” in Fig.3, Fig.3 legends, and the text.

- Please note that the exact p values are not provided in the legends of figures 1a-d, f, h-l; 2a-f, i; 3d-f, h, m; 4a-f; 5e, g-h; 7a-c, e-f; 8c-e, g-i; EV 1b-d, f-h.

Response: We have replaced asterisks with exact p values in these figures.

- Please define the error bars in the legend of figure 1b.

Response: We have defined the error bars in Fig. 1b in the figure legend.

- Please remove the sentence 'Source data are available online for this figure' from the legends. SD will be directly linked to the figures in the online version of the paper, thus this will be obvious.

Response: We have removed the sentence from all figure legends.

- Please add scale bars of similar style and thickness to all microscopic images (main, EV and Appendix figures), using clearly visible black or white bars (depending on the background). Please place these in the lower right corner of the images themselves. Please do not write on or near the bars in the image but define the size in the respective figure legend. Presently, most scale bars are too thin. Please improve.

Response: We have modified scale bars in all figures accordingly.

- Please make sure that all the funding information is also entered into the online submission system and that it is complete and similar to the one in the acknowledgement section of the manuscript text file. Presently, the grant from the Natural Science Foundation of Zhejiang Province (LZ24H090003) is missing in the submission system. Please check.

Response: We have included the funding information for LZ24H090003 in the online submission system.

- Please make sure that all figure panels are called out separately and sequentially. It seems, presently a panel 3D and individual callouts for the panels of Fig. 4 are

missing. Please check.

Response: As requested, we have called out all the missing panels in the text.

- Please make sure that the Appendix figures are named Appendix Fig. Sx and called out using this name. There is one callout for "Supplementary Figure 2" in the manuscript text. Please check.

Response: We are sorry for the mistake. We have changed "Supplementary Figure 2" to "Appendix Fig. S2A" in the Methods part, line 346.

- There are three Excel files uploaded ZIPed together (RAW1_LPS IgG.xlsx, RAW2_Con YOD1 IP.xlsx, and RAW3_LPS YOD1.xlsx). I assume this refers to Appendix Table S1 ("Appendix Table S1 - The data are available online as EXCEL files"). Please upload these files as separate datasets using the filename Dataset EVx. Please put a legend for these on the first TAB. Moreover, please add callouts for these files to the manuscript text.

Response: As suggested, we have renamed these files as Dataset EV1-3. The legend was included in each file. We have also added callout for these files in the text, line 186.

- Please remove the present text regarding Appendix Table S1 from the Appendix ("Appendix Table S1 - The data are available online as EXCEL files."). As indicated above, these Excel files need to be uploaded as Datasets EV1, EV2 and EV3. Then rename the Appendix table accordingly, i.e. Appendix Table S2 will become S1 and Appendix Table S3 will become S2. Finally, please update their callouts.

Response: We have made changes according to instructions. The callouts for Appendix Table S1 and S2 are included in lines 466 and 487, respectively.

- Thank you for providing the requested source data. Please upload this as one folder per main figure (with all files for one figure in one folder and ZIPed) and one folder for EV and Appendix figure.

Response: As requested, we have uploaded source data as one folder per figure.

- Please note that all corresponding authors are required to supply an ORCID ID for their name upon submission of a revised manuscript. Please do that for co-corresponding author Jing Ruan. Please find instructions on how to link the ORCID ID to the account in our manuscript tracking system in our Author guidelines: <http://www.embopress.org/page/journal/14693178/authorguide#authorshipguidelines>

Response: As requested, we have included ORCID ID for Jing Ruan in the online submission system.

In addition, I would need from you uploaded separately:

- a short, two-sentence summary of the manuscript (not more than 35 words).
- two to four short (!) bullet points highlighting the key findings of your study (two

lines each).

- a schematic summary figure as separate file that provides a sketch of the major findings (not a data image) in jpeg or tiff format (with the exact width of 550 pixels and a height of not more than 400 pixels) that can be used as a visual synopsis on our website.

Response: As requested, we have provided a short summary, bullet points, and the visual synopsis figure.

Best,

Referee #1:

The authors have addressed most of my concerns. I do think there needs to be a paragraph in the discussion concerning how YOD1 has been shown to be involved in IL-1 and LPS (TRAF6) and why the authors think they are not seeing the same results. Otherwise, I think the concerns have been addressed.

Response: As suggested, we have discussed the discrepancy in the discussion, lines 299-305.

Referee #2:

In this revised manuscript, Shen et al. have effectively addressed the questions raised during the review of their study, which identifies YOD1 as a novel regulator of colitis via the NOD2 signalling pathway. The manuscript has undergone substantial modifications, resulting in significant improvements. The key strengths of the revision include:

1. Enhanced histological analysis from in vivo studies, incorporating both mouse and patient data.
2. Clearer delineation of macrophages as a driving factor in the observed phenotype.
3. Inclusion of additional controls to rule out pathway cross-talk, ensuring that the YOD1 phenotype is attributed to NOD2 rather than other pathways like LPS/TNF.
4. Impressive immunoprecipitation of the NOD2 complex, demonstrating that RIPK2

levels are affected by YOD1 activity, rather than the complex formation itself in response to NOD2 stimuli.

5. Improved biochemical analysis of the increased K48 ubiquitination of RIPK2 in the absence of YOD1 under denatured conditions.

6. Effective use of RIPK2 knockdown via AAV, showing that RIPK2 knockdown phenocopies the YOD1 knockout.

Overall, these revisions strengthen the study, making it more robust. I am confident that the authors have successfully identified and validated an important regulator of the NOD2 signalling pathway, which may hold potential for therapeutic intervention.

Referee #3:

In their revised version, the authors of the manuscript "YOD1 sustains NOD2-mediated protective effects in colitis by stabilizing RIPK2" have addressed most of the initial concerns raised by this reviewer. There are however some minor corrections that should be addressed prior to publication.

Figure 2

2G: The F4/80 staining in the Yod1^{-/-} DSS condition does not look convincing given that the zoomed image amplifies an area of stain that is in the lumen of the colonic roll. This staining looks to be more like background than a real stain. Given that intestinal immune cells are mostly found in the lamina propria, the authors should zoom into an area where the viewer can see the lamina propria and a somewhat intact tissue architecture (as they have done in the Yod1^{-/-} condition).

Response: We have included zoomed images showing F4/80 staining in the lamina propria in Fig. 2G.

2H: Please include a representative gating scheme for the flow cytometry contour plots.

Response: As suggested, we have included the representative gating scheme in the Appendix Fig. S5.

2I: 1) The Y-axis is confusing. Is this a percentage of all cells in the colon? Viable cells? Singlets? Myeloid cells? Please clarify. Furthermore, including a representative gating scheme in the supplementary information as previously suggested also clarifies these concerns. 2) Spelling... "Neutrophils" not "Neutrophiles". Please correct.

Response:

(1) We are sorry for this confusion. We have changed the Y-axis of Fig. 2I to clearly show that the Y-axis indicates percentages of cell populations in CD45⁺ cells. We have also shown the gating strategy in the Appendix Fig. S5.

(2) We thank the reviewer for pointing out the mistake, which has been corrected.

Prof. Xu Wang
Wenzhou Medical University
Chashan High Education Park
Oujiang Laboratory
Zhejiang 325035
China

Dear Prof. Wang,

I am very pleased to accept your manuscript for publication in the next available issue of EMBO reports. Thank you for your contribution to our journal.

Yours sincerely,
